# Constraining composition and temperature variations in the mantle transition zone

Wen-Yi Zhou [1,2✉], Ming Hao [1,2], Jin S. Zhang [1,2✉], Bin Chen [3], Ruijia Wang [1] & Brandon Schmandt[1]

The mantle transition zone connects two major layers of Earth's interior that may be compositionally distinct: the upper mantle and the lower mantle. Wadsleyite is a major mineral in the upper mantle transition zone. Here, we measure the single-crystal elastic properties of hydrous Fe-bearing wadsleyite at high pressure-temperature conditions by Brillouin spectroscopy. Our results are then used to model the global distribution of wadsleyite proportion, temperature, and water content in the upper mantle transition zone by integrating mineral physics data with global seismic observations. Our models show that the upper mantle transition zone near subducted slabs is relatively cold, enriched in wadsleyite, and slightly more hydrated compared to regions where plumes are expected. This study provides direct evidence for the thermochemical heterogeneities in the upper mantle transition zone which is important for understanding the material exchange processes between the upper and lower mantle.

[1] Department of Earth and Planetary Sciences, University of New Mexico, Albuquerque, NM, USA. [2] Institute of Meteoritics, University of New Mexico, Albuquerque, NM, USA. [3] Hawaii Institute of Geophysics and Planetology, University of Hawai'i at Mānoa, Honolulu, HI, USA. ✉email: zwy1993@unm.edu; jinzhang@unm.edu

The chemical composition of bulk silicate Earth is highly controversial, ranging from the Si-poor pyrolite (olivine content 53–65%) to the Si-rich chondritic or perovskitic composition (olivine content 30–45%)[1–4]. Although the pyrolitic upper mantle is less debated, growing evidence suggests that the lower mantle can be chemically more enriched in Si, that is bridgmanite[2,5–8]. As a result, the silicate Earth is likely to be compositionally layered. This view, however, has been challenged by the widely observed material exchange between the upper and lower mantle suggested by seismic tomography, such as slabs penetrating into the lower mantle[9], and possible plumes rising beneath hotspots[10]. It is possible that such wide-range material exchange did not happen until recently in Earth's history and the time is too short to completely homogenize the lower mantle[11,12]. As the mantle transition zone (MTZ) connects the upper and lower mantle, mapping the thermochemical heterogeneities in MTZ is critical for understanding the material exchange between the upper and lower mantle, as well as volatile storage and transport in the deep Earth.

Unlike the low water solubility of the major upper and lower mantle minerals, the most abundant mineral phases in the MTZ, wadsleyite and ringwoodite, can store up to 2–3 wt% water as hydroxyl groups in their crystal structures[13]. Although later studies suggested that the upper limit of water concentration in wadsleyite and ringwoodite is ~1 wt% under normal mantle geotherm[14], the MTZ is still a giant potential water reservoir in the deep Earth. The possible enrichment of water in the MTZ can affect not only the seismic wave speeds of minerals[15–17], but also the phase transition boundary and binary phase loop width of relevant phase transitions in the MTZ pressure-temperature (P-T) space[18,19].

The depth, width, and magnitude of the 410, 520, and 660 km seismic discontinuities (hereinafter referred to as the 410, 520, and 660), which are mainly caused by the olivine to wadsleyite, wadsleyite to ringwoodite, and ringwoodite to bridgmanite +ferropericlase transitions, are functions of not only $T$[20], but also chemical composition, such as water[18] and Fe#[20]. Prior attempts have been made utilizing mineral physics data to better understand the seismic observations of these discontinuities. For example, Houser[21] used the 410 and 660 depth and $V$s anomaly in the lower MTZ to map the global water distribution assuming a pyrolitic MTZ. Wang et al.[15] took one step further by taking the $\Delta T$ which is temperature anomaly into account, using both the 660 depth and $V$s anomaly in the lower MTZ to map the water distribution again assuming a pyrolitic MTZ, although at least 20% of the lower MTZ cannot be explained by the pyrolitic composition within the reasonable water content and $\Delta T$ range assumed in Wang et al.[15]. As suggested in Zhang et al.[22], it is possible that chemical composition in the MTZ, at least locally, can significantly deviate from the idealized pyrolite model and influence seismic observations of discontinuities. Therefore, modeling the water content, $\Delta T$, and the bulk mineralogical composition together is needed to quantify thermochemical heterogeneities in the MTZ.

The most important physical properties needed for such modeling work are the high $P$-$T$ elastic properties of wadsleyite and majoritic garnet in the MTZ. Garnet (pyrope-grossular-almandine-majorite solid solution) is among the most intensively studied mantle minerals with numerous high $P$-$T$ sound velocity measurements available[23,24]. For example, the sound velocities of a synthetic garnet with "pyrolite minus olivine" composition were determined up to 18 GPa and 1673 K[4] and a polycrystalline almandine sample has also been measured up to 19 GPa and 1700 K[24]. Previous sound velocity measurements on wadsleyite have covered a wide range of Fe# (0–11.2) and water content (0–2.9 wt%) at high-$P$ ambient-$T$ or ambient $P$-$T$

conditions[17,25–30]. In particular, Zhou et al.[29] has established a Fe# and water content dependent single-crystal elasticity model of wadsleyite based on their own experimental data and all previous measurements[17,25–30]. However, the velocities of hydrous Fe-bearing wadsleyite samples have never been measured at simultaneously high $P$-$T$ conditions, and previous ultrasound interferometry experiments on polycrystalline wadsleyite were conducted outside the stability field of wadsleyite[31–33]. Moreover, the single-crystal elastic moduli ($C_{ij}$s) of wadsleyite have never been determined at simultaneously high $P$-$T$ conditions. Fe# and water content both have moderate effects on the $P$ dependence of the $C_{ij}$s, bulk modulus ($Ks$), and shear modulus ($G$) at 300 K (ref. [29] and references cited in ref. [29]); however, whether these effects are still significant at high $T$ remains unknown. First-principles calculations have provided useful insights toward the $P$ and $T$ dependence of the $C_{ij}$s of wadsleyite[16]. Given the large discrepancy between the computationally and experimentally determined elasticity data of ringwoodite[15], it is worthwhile to examine the single-crystal elasticity of wadsleyite experimentally in the high $P$-$T$ space. Therefore, in this study, we measured the sound velocities of the hydrous Fe-bearing wadsleyite single crystals (Fe# = 9.4 (2), 0.15 (4) wt% water) up to 16 GPa and 700 K using Brillouin spectroscopy (Fig. 1 and Supplementary Figs. 1–3).

Compared with the more complicated 520 and 660 discontinuities, which may be affected by other phase transitions nearby (e.g., exsolution of Ca silicate perovskite, majoritic garnet-bridgmanite transition), the 410 is solely related to the olivine-wadsleyite phase transition.

In this study, we exploit the petrological simplicity of the 410. By integrating the experimental constraints on the high $P$-$T$ elasticity of wadsleyite and majoritic garnet along with previously published mineral physics and global seismic imaging results, we map the global distributions of wadsleyite proportion, $\Delta T$, and water content in the upper MTZ, and statistically analyze their associations with different geographic and tectonic settings.

## Results and discussion

**High $P$-$T$ single-crystal elasticity of wadsleyite.** Using high $P$-$T$ diamond anvil cell combined with Brillouin spectroscopy method, we measured sound velocity on three wadsleyite platelets (Fe# = 9.4 (2), 0.15 (4) wt% water) with different orientations up to 16 GPa and 700 K (Fig. 1). We then obtained the best-fit $P$ and $T$ dependence of $Ks$, $G$, and $C_{ij}$s using $T$ dependent finite strain equations of state (see Methods). The aggregate elastic properties of the wadsleyite used in this study are: density $\rho_0 = 3.595$ (6) g cm$^{-3}$, $Ks_0 = 165$ (2) GPa[29], $G_0 = 104$ (2) GPa[29], $(\partial Ks/\partial P)_{T0} = 5.1$ (1), $(\partial^2 Ks/\partial P^2)_{T0} = -0.14$ (2) GPa$^{-1}$, $(\partial G/\partial P)_{T0} = 1.9$ (1), $(\partial^2 G/\partial P^2)_{T0} = -0.07$ (1) GPa$^{-1}$, $(\partial Ks/\partial T)_{P0} = -0.018$ (2) GPa K$^{-1}$, and $(\partial G/\partial T)_{P0} = -0.014$ (1) GPa K$^{-1}$ (Supplementary Table 1). As shown in Supplementary Table 2, the $(\partial Ks/\partial T)_{P0}$ and $(\partial G/\partial T)_{P0}$ of wadsleyite measured in this study agree with what were given in Gwanmesia et al.[33] and Isaak et al.[34], suggesting that the effects of water and Fe on the $T$ derivatives of the elastic moduli are negligible. The $(\partial C_{ij}s/\partial T)_{P0}$ for our wadsleyite sample are shown in Supplementary Table 3. The six diagonal $C_{ij}$s are more sensitive to $T$ compared with the off-diagonal $C_{ij}$s (Supplementary Fig. 3). $T$ also has a more pronounced effect on $V$s than $V$p.

**Mineral physics constraints on the upper MTZ seismic observations.** Based on the $(\partial Ks/\partial T)_{P0}$ and $(\partial G/\partial T)_{P0}$ determined in this study and the relationship between water content and $Ks_0$, $G_0$, $(\partial Ks/\partial P)_{T0}$, and $(\partial G/\partial P)_{T0}$ derived in Zhou et al.[29], we found that the hydration effect diminishes with $P$ but slightly

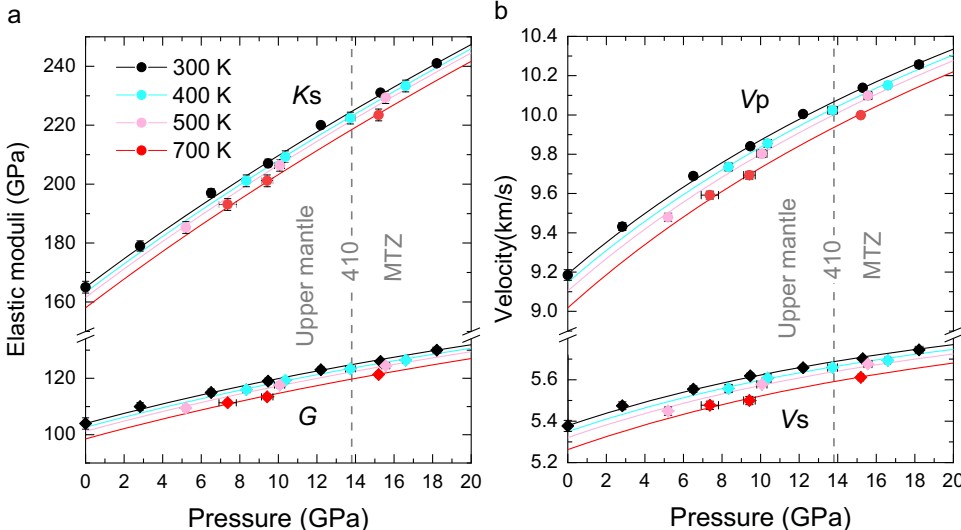

**Fig. 1 High *P-T* elastic properties of the hydrous Fe-bearing wadsleyite sample used in this study. a** Bulk modulus *K*s and shear modulus *G* of the isotropic polycrystalline wadsleyite as a function of *P* and *T*. **b** *V*p and *V*s of the isotropic polycrystalline wadsleyite as a function of *P* and *T*. Solid lines represent fourth-order finite strain equation of state fitting results. Voigt–Reuss–Hill averaging scheme is used to calculate the *K*s, *G*, *V*p, and *V*s of the isotropic polycrystalline wadsleyite at 300, 400, 500, and 700 K, respectively. Data along the 300 K isotherm are adopted from Zhou et al.[29]. Error bars smaller than the symbols are not shown.

enhances with *T*. Under high *P-T* conditions of the upper MTZ, adding 1 wt% water in wadsleyite decreases its *V*p by ~1% and *V*s by ~0.5% (Fig. 2). In addition, the *V*p and *V*s decrease caused by adding 1 wt% water in wadsleyite is comparable to the velocity reduction resulted from *T* increase of 300 and 100 K, respectively (Fig. 2).

In addition to *T* and water content, wadsleyite proportion also affects the *V*p and *V*s of upper MTZ. Pyroxenes completely dissolve into majoritic garnet at depths shallower than 450 km[35]. As suggested by previous phase equilibrium experiments (e.g., Ishii et al.[35,36]), wadsleyite and majoritic garnet are the only major minerals at depths greater than 450 km in the upper MTZ. Therefore, the upper MTZ can be viewed as a simplified binary mineral mixture of wadsleyite and majoritic garnet, although it does not cover the full range of possible compositional variations in the MTZ since we ignored the possible existence of other minor mineral phases (e.g., clinopyroxene between 410 and 450 km depth). The lateral variations of wadsleyite and majoritic garnet contents in the upper MTZ are directly linked to the change of major element distributions in the MTZ, including Fe, Al, Ca, Si, etc. For example, a wadsleyite-poor region in the upper MTZ is expected to be Fe, Al, Ca, and Si-rich, but Mg-poor[35,36].

Wadsleyite is ~4% faster than majoritic garnet in both pyrolite and harzburgite bulk compositions under MTZ *P-T* conditions (the thermoelastic parameters of wadsleyite and garnet used for calculating the high *P-T* velocities are shown in Supplementary Tables 2, 4), therefore increased wadsleyite proportion would result in higher *V*p and *V*s in the upper MTZ.

Seismically fast and slow anomalies have been widely identified in the upper MTZ, e.g., positive anomalies beneath the western Pacific subduction zone, negative anomalies under the central Pacific plate[37]. The 410 depth, influenced by both Δ*T* and water content (see Methods), also varies globally[38]. The effect of $Fe^{3+}$ content on the 410 depth is expected to be very small considering the small Fe# range near 410 km depth[20]. The existence of $Fe^{3+}$ may broaden the 410[18], while accurate quantification suffers from the limited experimental data as well as the unknown redox state near 410 km depth in the Earth's interior. Therefore, we did not consider the effect of $Fe^{3+}$, only

water content and wadsleyite proportion were considered as the compositional factors in our modeling.

**Modeling the composition and *T* variations in the upper MTZ.** We attempt to constrain the global variations of wadsleyite proportion, water concentration, and Δ*T* in the upper MTZ from the seismically determined *V*p and *V*s at 450 km from the SP12RTS model[37], and the 410 depths from Huang et al.[38]. The SP12RTS tomography models used the inversion framework of the S40RTS[39] *V*s tomography model. The most recent global MTZ discontinuity topography model by Huang et al.[38] also used the S40RTS tomography model to correct for the lateral seismic heterogeneities in the upper mantle. Therefore, the seismic data products we chose in this study are internally consistent.

A grid search method was employed to calculate the predicted *V*p and *V*s anomaly at 450 km depth and the 410 depth by allowing the wadsleyite proportion, Δ*T*, and water content to vary between 13 to 93 vol%, −300 to 300 K, 0 to 2 wt%, respectively, based on the mineral physics data in this and previous studies (Supplementary Tables 2, 4) (Eqs. 1–4 in Methods). We then compared the mineral physics predicted values with the internally consistent global seismic models presented in Huang et al.[38] and Koelemeijer et al.[37] to search for the mineral physics models with misfits less than 1.0 and 1.5 defined by Eq. 5 in Methods. The best-fit wadsleyite proportion, Δ*T*, and water content models were calculated by averaging those selected mineral physics models with misfits less than 1.0 and 1.5 (Fig. 3 and S4). Uncertainties involved in the modeling are considered and evaluated (See Methods and Supplementary Discussion 1).

The average of the best-fit models with misfit <1.0 yields an estimated global average wadsleyite content of 56 ± 6 vol%, *T* of 1866 ± 42 K, and water content of 0.28 ± 0.08 wt%. The results we obtained from the best-fit models with misfit <1.5 yields slightly higher average water content (0.40 ± 0.07 wt%). However, the lateral variations for water content, Δ*T*, and wadsleyite proportion in the upper MTZ remain consistent and the spatial distribution is stable as shown in Fig. 3 (misfit <1) and Supplementary Fig. 4 (misfit <1.5).

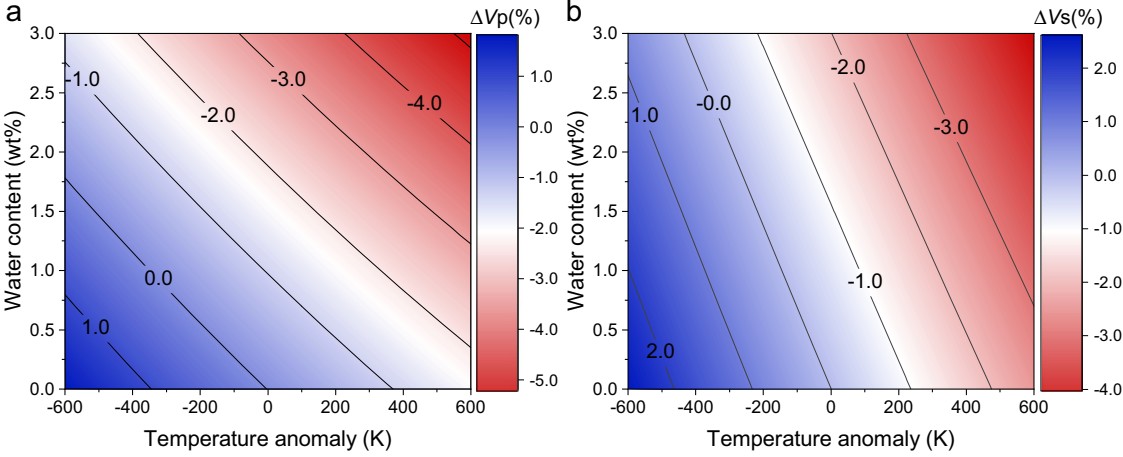

**Fig. 2 Effects of water content in wadsleyite and temperature anomaly ΔT on the sound velocities of wadsleyite. a** Relative variation, in percent, of wadsleyite $V_p$ ($\Delta V_p$) as a consequence of water content and $\Delta T$ at the P-T conditions near the 410. **b** Relative variation, in percent, of wadsleyite $V_s$ ($\Delta V_s$) as a consequence of water content and $\Delta T$ at the P-T conditions near the 410.

In the maps derived from best-fit models (Fig. 3 and S4), the maximum wadsleyite proportion, $T$, and water content differences for the upper MTZ are ~35 vol%, ~270 K, and ~0.5 wt %, respectively. To statistically analyze their associations with different geographic and tectonic settings, we selected four regional groups: ocean, continents, subducting slabs, and hotspots (plumes).

The wadsleyite content distribution map is shown in Fig. 3a. The most prominent feature is the strong correlation between the high wadsleyite content and the places where slabs are expected, e.g., beneath the circum-Pacific subduction zones (Figs. 3a, 4). A possible explanation can be the olivine-rich harzburgitic lithospheric mantle of the slabs penetrating or accumulating in the MTZ[9]. For example, seismic tomography has revealed possible slab stagnation in the MTZ under northeastern China[40], Bonin arc[9], eastern Java subduction zone[9], southern Chile[41], etc. Some locations where slabs penetrate through the MTZ also coincide with the regions with high wadsleyite content, such as beneath northern New Zealand[9]. In addition, due to the existence of subducting slabs under most continents with active continental margins (e.g., western South and North America, East Asia), higher wadsleyite content was found beneath continents compared with oceans (Fig. 4).

Another interesting correlation is found between the regions depleted in wadsleyite content and the deeply sourced hotspots[10] (Fig. 3a). Under 18 out of 22 deeply sourced hotspots, the wadsleyite content is lower than what is expected in a pyrolite model (Fig. 3a). As shown in Fig. 4a, the average upper MTZ beneath all deeply sourced hotspots has ~ 2 vol% less wadsleyite than the global average MTZ (~56 vol%), and ~10 vol% less wadsleyite than the average upper MTZ near slabs. This observation could be explained by the upwelling of the possible Si-rich, thus "olivine-poor" material from the lower mantle associated with the plumes. Although still controversial, recent sound velocity, density, and viscosity experiments on lower mantle minerals support a Si-rich bridgmanite-dominated lower mantle[2,5–8]. After being transported to the upper MTZ by the rising plumes, the bridgmanite-rich lower mantle materials would transform into a relatively wadsleyite-poor aggregate, and eventually result in the low wadsleyite content under these deeply sourced hotspots. Our observation is also consistent with recent geodynamical modeling results, which suggest the compositionally "garnet"-rich, therefore "olivine"-depleted plumes can better reproduce the expected plume features such as tail size, stability, and buoyancy[42].

$\Delta T$ distribution in the upper MTZ is shown in Fig. 3c. The most pronounced feature in this figure is the unusually high $T$ beneath the Pacific Ocean (Fig. 3c), consistent with a previous $T$ model derived from $V_s$ only at 300 km[43]. The high $T$ beneath the Pacific Ocean in the MTZ and upper mantle may be related to the relatively large number (~60%) of the deeply originated hotspots found in this region[10] (Fig. 4).

On the other hand, the upper MTZ under continents is the coldest among the four regional groups (Fig. 4), which may be expected due to the existence of the thick and cold lithosphere under continents[44]. Lower $T$ is also observed beneath some subduction zones, e.g., beneath southeastern Asia (Fig. 3c). Interestingly, the $T$ under the Cascadia subduction zone is actually high, which can be explained by the slow subduction (~3 cm/year) of the young (<10 Ma) Juan De Fuca slab with a thick sediment cover[45], as well as by the nearby Yellowstone hotspot in the Western United States (Fig. 3c). Due to the different thermal states of different subducting slabs, the $T$ under subduction zones is highly variable (Fig. 4b).

One mysterious feature shown in Fig. 3c is the low $T$ beneath the Red Sea near where a plume is expected to its southeast. The global topography model of the 410 adopted in this study suggested an uplifted 410 beneath the Red Sea[38], likely associated with a negative $\Delta T$. On the contrary, a local seismic study near the Red Sea reported a depressed 410 up to 30 km[46], suggesting the MTZ beneath the Red Sea is hotter than the ambient mantle by up to 280 K. More seismic studies with denser ray coverage are needed to clarify the 410 depth variation in this region.

Although the absolute water content in the MTZ is not well resolved, the lateral variation of water content in the upper MTZ is stable for different ensemble sizes from the grid search as shown in Fig. 3e (misfit <1) and Supplementary Fig. 4e (misfit <1.5).

The circum-Pacific regions show a slightly elevated water content, consistent with the existence of subduction slabs. Pore fluids or hydrous minerals (e.g., antigorite, Ice VII, phase A) carried by fast-subducting cold slabs could supply water to the MTZ[47]. The MTZ beneath the Western Pacific subduction zone is suggested to be hydrated by seismic observations[48], electrical conductivity studies[49,50], and basalt geochemistry[51]. Higher water content in the upper MTZ under the Western United States seen in Fig. 3e is also supported by the observed low-velocity layer atop the 410[52,53] and high seismic attenuation[54]. It can be caused by the dehydration of the old stagnant Farallon slab[53] or the

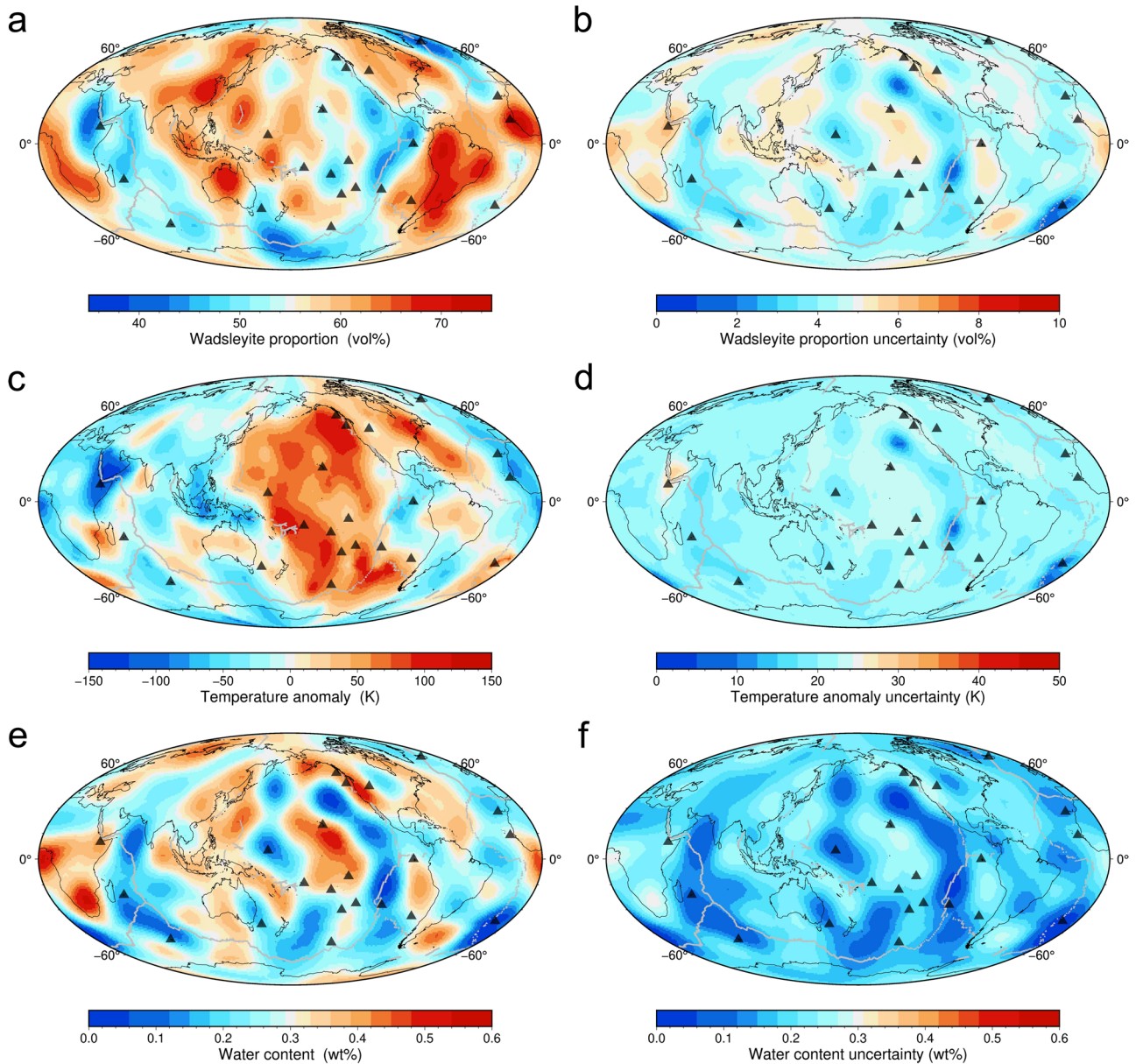

**Fig. 3 Wadsleyite proportion, temperature anomaly $\Delta T$, and water content in the upper MTZ constrained from the depth variation of the 410, $V_p$, and $V_s$ anomaly. a** Global distribution of wadsleyite proportion. **b** Global distribution of the uncertainty of wadsleyite proportion. **c** Global distribution of $\Delta T$. **d** Global distribution of the uncertainty of $\Delta T$. **e** Global distribution of water content. **f** Global distribution of the uncertainty of water content. Deeply sourced hotspots[10] are plotted as black triangles.

sediments atop the Juan de Fuca slab, which is the north remnant of the Farallon slab[45].

Recent studies suggested that the large low-shear-velocity provinces (LLSVPs) in the lower mantle may be locally hydrated by fluids released from the phase δ-H solid solution $(AlO_2H\text{-}MgSiO_4H_2)$[55] and pyrite $FeO_2H_x$[56]. Upwelling plumes originating from the hydrated regions in LLSVPs may hydrate the upper MTZ locally, as seen under Hawaii and South Africa. On the other hand, the Archean and Paleoproterozoic cratons in southern Africa were formed and welded by processes similar to modern-day subduction[57], so the MTZ under southern Africa may also have been hydrated by ancient subducting slabs[58].

Other hydrous or anhydrous areas are interspersed in the upper MTZ, possibly due to three-dimensional mantle convection[59]. The correlations between high (or low) water concentration and surface properties (i.e., lands, oceans, hotspots, or slabs) may be complicated by three-dimensional mantle convection that offsets deep thermal or compositional anomalies from their past or present surface expressions (Fig. 4c).

**Summary**. We found that the upper MTZ near subducted slabs is relatively cold, enriched in wadsleyite, and slightly more hydrated compared to regions where plumes are expected. It is very likely that both the subduction of Si-poor oceanic mantle lithosphere and upwelling of the possibly Si-rich lower mantle materials contribute to the large composition and $T$ heterogeneities in the upper MTZ. We would also like to point out that the results presented in this study are restricted by the mineral physics constraints and seismic resolution. As a result, some other factors, such as the presence of minor phases, fluids, or anelastic effects were not taken into account.

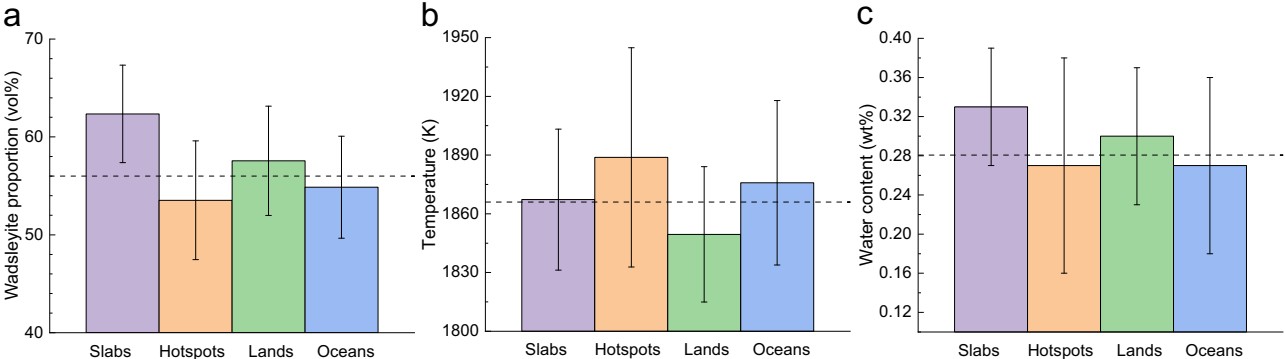

**Fig. 4 Wadsleyite proportion, temperature *T*, water content in the upper MTZ near slabs, beneath deeply sourced hotspots, lands, and oceans. a** The wadsleyite proportion difference between slabs and hotspots is greater than 1 standard deviation. **b** The ~30 K *T* differences between slabs and hotspots as well as lands and oceans are both slightly smaller than 1 standard deviation largely due to the different thermal states of the grouped individual locations used in this study. **c** The subtle difference in estimated water content is much smaller than the 1 standard deviation and the absolute values of the water content are only weakly constrained. The dashed lines represent the global average.

Extensive collaborations between seismologists and mineral physicists are needed to address these technical challenges in order to quantitatively understand the multidimensional thermo-chemical heterogeneities in the petrologically more complicated regions inside the Earth.

## Methods

**Sample characterization**. The samples used in this study are the same as the ones used in Zhou et al.[29] for ambient-*T* high-*P* sound velocity measurements. Hydrous wadsleyite single crystals were synthesized from San Carlos olivine powder at 1673 K and 16 GPa for 24 h in the 1000-ton Multi-Anvil Press at the University of Hawai'i at Mānoa. We used the JEOL 8200 Electron Microprobe (beam current = 20 nA; accelerating voltage = 15 kV) to determine the composition of the wadsleyite samples at the University of New Mexico (UNM). The electron microprobe analysis on 5 single-crystal grains yielded an average Fe# of 9.4 (2). The Mössbauer spectroscopy experiment was performed on ~80 randomly oriented wadsleyite grains in the offline Mössbauer spectroscopy lab at sector 3, Advanced Photon Source, Argonne National Laboratory. The $Fe^{3+}/\Sigma Fe$ of synthetic wadsleyite is determined to be 0.3. The Nicolet Nexus 670 Fourier Transformed Infrared Spectrometer (FTIR) at UNM was used to determine the water content using the calibration presented in Libowitzky and Rossman[60]. Using the calibration procedure outlined in Deon et al.[61] yielded a similar water content as shown in Supplementary Table 5. The technical details and results of the Electron microscope analysis, Mössbauer spectroscopy, and part of the FTIR experimental results were all presented in Zhou et al.[29]. In this study, for more precise water content determination, we polished another eight randomly selected wadsleyite single crystals for additional unpolarized FTIR measurements. The results are shown in Supplementary Fig. 5 and Supplementary Table 5. The updated water content is 0.15 (4) wt% by averaging the unpolarized IR absorption spectra on 17 crystals. Nine crystals have been measured and shown in Zhou et al.[29], and the rest are the new measurements conducted in this study. The new water content of 0.15 (4) wt% is about 0.01 wt% higher than the water content determined from nine crystals (0.14 (4) wt%) in Zhou et al.[29], which is within the water content uncertainty of 400 ppm (Supplementary Table 5). The small change of water content has a negligible effect on the composition-dependent single-crystal elasticity model presented in Zhou et al.[29].

**High *P-T* Brillouin spectroscopy experiments**. We performed high *P-T* Brillouin spectroscopy experiments on three wadsleyite single-crystal platelets with orientations: (−0.3420, 0.8718, 0.3506), (0.0748, 0.3154, −0.9460), and (0.5341, 0.0953, −0.8401). Each crystal (~50 μm × 50 μm × 12 μm) was loaded into a high-*T* BX90 diamond anvil cell (DAC) along with two ruby spheres for *P* determination. The pair of diamond anvils (350 μm culet) was oriented before experiments so that the slow directions of the top and bottom diamonds matched each other. A 250 μm hole was drilled in a pre-indented Rhenium gasket. Neon was gas-loaded into the sample chamber as a *P*-transmitting medium at GeoSoilEnviroCARS, Advanced Photon Source. High *T* was generated by Platinum resistive heaters located at the center of the DAC[62]. We attached two K-type thermocouples to the diamond surface as close as possible to the diamond culet for *T* measurement. The *T* difference given by two thermocouples is <15 K throughout all experiments.

We performed Brillouin spectroscopy experiments at the Laser Spectroscopy Laboratory at UNM using a 532 nm single-mode diode-pumped solid-state laser as the light source. We used 50° symmetric forward scattering geometry and a

standard silica glass Corning 7980 was employed to calibrate the scattering angle every two months[63]. The high *P-T* Brillouin measurements of wadsleyite crystals lasted for eight months, and the calibrated scattering angles used for data analysis are 50.42 (5)°, 50.60 (8)°, 50.68 (6)°, and 50.77 (6)°. The Brillouin frequency shift was measured using a six-pass tandem Fabry-Pérot interferometer. At each *P-T* condition, *V*p and *V*s of every single crystal were measured from 0° to 360° with a step of 15° (24 different Chi angles). Typical Brillouin spectra are shown in Supplementary Fig. 1. Brillouin experiments under high *P-T* conditions along 400, 500, and 700 K isotherms were conducted for all three wadsleyite platelets.

Using the Christoffel equations, we inverted the best-fit $C_{ij}$s of wadsleyite based on the high *P-T* phonon direction-*V*p-*V*s dataset on the three wadsleyite single crystals (Supplementary Fig. 2) with initial estimated high *P-T* densities. The RMS error at each *P-T* condition is less than 60 m/s. We then utilized Voigt–Reuss–Hill averaging scheme[64] to calculate the aggregate *V*p and *V*s at each high *P-T* condition. It is worth noting that although the density is based on an initial guess, the *V*p and *V*s derived here are actually true values. With fixed ambient *P-T* $Ks_0$, $G_0$, and $\rho_0$ from Zhou et al.[29], thermal expansivity[65] $\alpha$ as $2.31(3) \times 10^{-5} + 1.18(3) \times 10^{-8}$ (*T* - 300) $K^{-1}$, we fitted the new high *P-T* velocity data collected in this study together with the room-*T* high-*P* elasticity data presented in Zhou et al.[29] using *T* dependent fourth-order finite strain equation of state[66,67], and derived the best-fit $(\partial Ks/\partial P)_{T0}$, $(\partial^2 Ks/\partial P^2)_{T0}$, $(\partial G/\partial P)_{T0}$, $(\partial^2 G/\partial P^2)_{T0}$, $(\partial Ks/\partial T)_{P0}$, $(\partial G/\partial T)_{P0}$, and densities. Then, we updated the $C_{ij}$s, $Ks$, $G$ at each high *P-T* condition with the derived true densities (Supplementary Table 1). Finally, using the aggregate thermoelastic parameters obtained above, we fitted *P-T*-$C_{ij}$s data to the *T*-dependent third-order or fourth-order finite strain equation of state[68] to obtain the best-fit $(\partial C_{ij}/\partial P)_{T0}$, $(\partial^2 C_{ij}/\partial P^2)_{T0}$, $(\partial C_{ij}/\partial T)_{P0}$, which were shown in Supplementary Table 3.

**Evaluation of water loss during the high *P-T* experiments**. We conducted the following three experiments to confirm that the wadsleyite crystals remained hydrated after the completion of the high *P-T* experiments: (1) We collected additional FTIR spectra on Wad34 at 300 K before heating, and at 300 K after it was heated to 400, 500, and 700 K (Supplementary Fig. 6). The water content is determined to be 1800 ppm before heating, and it remains the same after multiple heating and cooling cycles. There is no resolvable water loss. (2) We conducted additional Brillouin spectroscopy measurements on sample Wad9 (0.0748, 0.3154, −0.9460) at 400 K and 0 GPa after the completion of high *P-T* experiments on this sample. If significant water loss has taken place, the newly collected sound velocity data would be faster than the $C_{ij}$ model-predicted *V*p and *V*s values (Supplementary Fig. 7), which is not the case. (3) After we completed the 700 K and 9.4 GPa Brillouin spectroscopy experiments of wadsleyite sample Wad6 (−0.3420, 0.8718, 0.3506) and cooled it down to 300 K, we collected five additional Brillouin spectra of this sample at 300 K and 9 GPa along five additional different crystallographic directions. Due to the short collection time, the spectrum quality only allows reliable determination of *V*s at these five different directions. The obtained *V*s values nicely matched the $C_{ij}$ model-predicted *V*s values at 300 K and 9 GPa, as well as the *V*s at 300 K and 9.5 GPa measured before the sample was heated (Supplementary Fig. 8). Thus, the crystals are unlikely dehydrated during the high *P-T* experiments.

**Grid search modeling**. Based on the experimentally determined mineral physics data, we calculated the predicted *V*p and *V*s anomaly at 450 km depth and the 410 depth at each location by varying the wadsleyite content (13–93 vol % in 2 vol%

steps), $\Delta T$ (−300 to 300 K in 15 K steps), the water content in wadsleyite (0–2 wt% in 0.05 wt% steps), resulting in ~70,000 predicted models.

The $Vp$ anomaly compared with a dry pyrolite model at 450 km are estimated using the following equation:

$$\Delta Vp\_pred = (a * P_{Wad} + b * (100 - P_{Wad})) * 0.01 * \Delta T \\ + c * (P_{Wad} - 53) - d * W * P_{Wad} * 0.01 \quad (1)$$

where $P_{Wad}$ (vol%) is the wadsleyite content, $\Delta T$ (K) is the $T$ anomaly compared with 1870 K[19], and $W$ (wt%) is the water content in wadsleyite. $a$ (−0.0027) and $b$ (−0.0028) are the influence of $\Delta T = 1$ K on $\Delta Vp\_pred$ (%) of wadsleyite and majoritic garnet in the upper MTZ, respectively. $c$ (0.0397) refers to the influence of increasing wadsleyite proportion by 1% on $\Delta Vp\_pred$. The reference pyrolite model and harzburgite model we used in this study have 53 and 73 vol% wadsleyite at ~15 GPa 1870 K, according to the most recent experimental study by Ishii et al.[35,36]. The chemical compositions of relevant mineral phases in pyrolite and harzburgite were also adopted from refs. [35,36]. $d$ (0.9750) describes the relative change of $\Delta Vp\_pred$ of wadsleyite caused by adding 1 wt% water in wadsleyite (this study and ref. [29]) at the upper MTZ $P$-$T$ conditions. The parameters used to determine these coefficients ($a$, $b$, $c$, $d$) are reported in Supplementary Tables 2, 4.

Similarly, the $Vs$ anomaly change as a function of $P_{Wad}$ (vol%), $\Delta T$ (K), and $W$ (wt%) is calculated as:

$$\Delta Vs\_pred = (A * P_{Wad} + B * (100 - P_{Wad})) * 0.01 * \Delta T + C * (P_{Wad} - 53) \\ - W * P_{Wad} * 0.01 * D \quad (2)$$

with $A = -0.0040$ (this study), $B = -0.0045$, $C = 0.0868$, $D = 0.4850$. The parameters used to determine these coefficients ($A$, $B$, $C$, $D$) are reported in Supplementary Tables 2, 4. Both Eqs. 1 and 2 used volume-weighted velocities instead of the volume-weighted elastic moduli and density for calculating the velocities of garnet/wadsleyite mixtures due to the following two reasons: (1) Velocities of pyrolite calculated using these two approaches are very similar (the difference is <0.1%), as shown in Supplementary Table 6; (2) Avoid additional parameters related to $T$ and composition-dependent density in the modeling and reduce the total number of parameters.

The 410 depth is affected by both $\Delta T$ (K) and $W$ (wt%), and it is modeled as:

$$d410\_pred = \Delta T * e + \Delta T * \frac{W}{f} (\Delta T < 0) \quad (3)$$

$$d410\_pred = \Delta T * e (\Delta T \geq 0) \quad (4)$$

The effect of $\Delta T$ on the 410 depth is represented by $e$, which is determined to be 0.107 km K$^{-1}$ by Katsura et al.[20]. As suggested by Frost and Dolejš[18], increasing the water concentration of wadsleyite elevates the 410 discontinuity when $T$ is low, however, such effect vanishes at higher $T$. In this study, we assume a linear decrease of the 410 depth ($\frac{-\Delta T}{f}$; $f = 20$) as a function of $\Delta T$ by adding 1 wt% water in wadsleyite when the $T$ is lower than the reference $T$ of 1870 K.

We compared the values predicted from the grid search process in the $P_{Wad}$ - $\Delta T$ (K) - $W$ (wt%) data space with global seismic models (the 410 depths ($d410\_obs$) from ref. [38], $Vp$ and $Vs$ anomaly ($\Delta Vp\_obs$, $\Delta Vs\_obs$) at 450 km depth from ref. [37]) at each location. The goodness of the model at each location is evaluated based on all three seismic observations:

$$\text{Misfit} = \sqrt{\frac{(d410\_pred - d410\_obs)^2}{(d410\_obs\_error)^2} + \frac{(\Delta Vp\_pred - \Delta Vp\_obs)^2}{(\Delta Vp\_obs\_error)^2} + \frac{(\Delta Vs\_pred - \Delta Vs\_obs)^2}{(\Delta Vs\_obs\_error)^2}} \quad (5)$$

with $d410\_obs\_error = 1.3\%$[38]; $\Delta Vp\_obs\_error = 0.5\%$; $\Delta Vs\_obs\_error = 0.85\%$[69]. We also conducted additional tests to verify the robustness of our modeling results given the uncertainties of the mineral physics predictions involved in the modeling. Details can be found in Supplementary Discussion 1, Supplementary Fig. 9, and Supplementary Table 7.

We then picked the best-fit model with misfit <1 at each location and then averaged the wadsleyite proportion, $\Delta T$, and water content in wadsleyite of the chosen best-fit models (Fig. 3 and Supplementary Data 1). The standard deviation of these parameters in the chosen best-fit models would be the uncertainties for each parameter. We also averaged the best-fit models with misfit <1.5 at each location and the results are shown in Supplementary Fig. 4 and Supplementary Data 2. Due to the similar density of wadsleyite and majoritic garnet at the condition of the upper MTZ, the difference between wadsleyite vol% and wt% is less than 1% in the upper MTZ. Therefore, the water content in the upper MTZ at each location is derived by multiplying the wadsleyite proportion (vol%) and water content in wadsleyite (wt%). All the modeling codes are written in Python. All figures are plotted with PyGMT[70].

**Regional features of the upper MTZ.** To statistically analyze the features in the upper MTZ under different geographic and tectonic settings, we selected four regional groups in this study: ocean, continents, subducting slabs, and deeply sourced hotspots (plumes), and calculated their average wadsleyite proportion,

$\Delta T$, and water content in the upper MTZ. Slab locations near the upper MTZ are defined by software Slab2 and shown in Supplementary Fig. 10. The land and ocean locations are downloaded from the GMT database and shown in Supplementary Fig. 11. The locations for deeply sourced hotspots (plumes) used in this study meet ≥2 requirements for deeply sourced hotspots given by Courtillot et al.[10] and are shown in Fig. 3.

## Data availability
The authors declare that all data supporting the findings of this study are provided within the paper and the Supplementary Information files.

## Code availability
The python code is available by request from the corresponding author J.S.Z. and W.-Y.Z.

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

## Acknowledgements

This project is funded by NSF-EAR-1664471 (J.S.Z. and B.S.). We thank Dr. Quancheng Huang and Dr. Takayuki Ishii for providing data from their previous publications. B.C. acknowledges the support from NSF-EAR-1555388 and NSF-EAR-1829273. J.S.Z. is also supported by NSF-EAR-1847707. Use of the APS was supported by the US Department of Energy, Office of Science, Office of Basic Energy Sciences, under Contract No.DE-AC02-06CH11357.

## Author contributions

J.S.Z., B.S., W.-Y.Z., and M.H. designed research. B.C. synthesized samples. W.-Y.Z., M.H., and J.S.Z. conducted experiments. W.-Y.Z., J.S.Z., B.S., and R.W. wrote the code and made figures and tables. W.-Y.Z. and J.S.Z. wrote the manuscript with input from all authors.

## Competing interests

The authors declare no competing interests.
