## [Peer Review File · Nature Communications]

REVIEWER COMMENTS

Reviewer #1 (Remarks to the Author):

This manuscript presents experimental results on the single-crystal elasticity of hydrous Fe-bearing wadsleyite at high P-T conditions for the first time. These results were then compared with Vp and Vs velocities from seismic tomographic models and a model for the depth variation of the 410-km discontinuity to place constraints on the geographical distribution of wadsleyite proportion, temperature anomaly, and water content in the upper mantle transition zone (MTZ). This is a very well written manuscript with robust experimental data. The model presented is neat and powerful to resolve the non-uniqueness associated with inversions. I thus would like to recommend the publication of this manuscript in Nature Communications. Below are some comments/suggestions to consider/clarify.

1. I think the manuscript needs some work to better quantify the uncertainties for the wadsleyite proportion, T anomaly, and water content. It seems to me the current uncertainties come from the parameter distributions during the grid search. However, there are other sources of uncertainties not considered in this procession. For example, the resulted temperature ($\sim 1870 \pm 150$) and water content ($\sim 0-0.6$ wt%) in the upper MTZ (Fig. 3) are much higher than the highest temperature (700 K) and water content (0.18wt%) in the experiments. This could result in some uncertainties during extrapolation and should be included in the final inversion results. In addition, The uncertainties in the seismic tomography models for absolute values of Vp/Vs and 410 km depth were not discussed. These could be another significant source of error, although they should not invalidate the main distribution feature obtained from the study.
2. Better add error bars or state the uncertainties for the moduli and pressures in Fig. 1, Fig. S2, and Fig. S3.
3. Fig. S1: It looks that there are two peaks for the Vp signal on the left hand side of the spectrum and two overlapping peaks for Vp on the right hand side. What is the likely cause for the peak splitting? How Vp is obtained based on a spectrum like this?
4. Line 112: It should be "adding 1 wt% water in wadsleyite" instead of "adding 1 wt% in wadsleyite".
5. Line 134: It should be "T anomaly" instead of "T".
6. Figure 3a2 and Figure S4a2: I believe the color bar is incorrect for these two figures. The uncertainty should not be as large as $\sim 55-65\%$. I suspect a 53% should be subtracted.

Reviewer #2 (Remarks to the Author):

This paper reports measurements of the single-crystal elastic properties of an H-bearing wadsleyite at high pressures and temperatures. The results are used to develop a model for the variation of wadsleyite fraction, H content, and temperature in the Earth's upper transition zone. The work builds on an earlier study by Zhou et al 2021 which reported measurements on the same samples at ambient temperature and high pressure (although this should be stated more clearly in the present paper).

The elasticity data appears to be of generally high quality and the measurements on hydrous wadsleyite at simultaneous high P-T conditions have not been reported before and this certainly warrants publication. The authors also apply their data in an attempt to simultaneously constrain wadsleyite content, hydrogen content, and temperature in the upper part of the transition zone by combing their data with elasticity data for garnet/majorite (the other major phase in the region) and comparing with seismic velocities at 450 km depth as well as the depth of the 410-km discontinuity. I find this part of the paper to be more speculative and now well constrained. As a result, I don't think the paper warrants publication in Nature Communications.

Major comments:

1. Characterizing the water content in the sample is crucial for this kind of work, and the authors mainly refer to their earlier paper (Zhou et al EPSL 21) for details of the water content determination. However, I don't feel that characterization in that paper is sufficient, and more details are needed here. Determination of water content by FTIR requires specific calibration which can vary significantly between different laboratories. The authors here apparently only use 1 calibration. Why was that calibration chosen and how much do the results change when using a different calibration? If the presence of Fe³⁺ complicates the calibration by IR, then another technique such as SIMS should be used to determine the H₂O content. Quantification of water in anisotropic minerals by IR spectroscopy requires consideration of orientation effects, which apparently was not done here as only unpolarized spectra were used. Finally, it should be noted that the speciation of the H-defect location could affect the measured properties. What can be said about this from the IR spectra? This is all important b/c without a very good calibration and some knowledge of H speciation, it is difficult to compare the present results with those of other studies and the literature becomes confused with incompletely characterized samples.

2. According to Figure 1, it looks to my eye that there is a bigger reduction in moduli and velocity from 500-700 K than from 300-500K. This is somewhat surprising especially as the authors assume a linear variation with temperature. Is that correct? A separate issue is whether there could be some change to the sample on heating, for example change in the amount or speciation of hydrogen. What was done to verify that the sample did not change significantly upon heating?

3. The authors attempt to constrain the wadsleyite fraction, water content, and temperature from seismic velocity data at 450 km depth and the depth of the 410-km discontinuity. In doing so, several assumptions or simplifications must be made. A major one is the assumption that chemical composition (Fe, Al, Ca, Na, etc) does not change laterally. This is unlikely to be true (see Schmerr and Garnero, 2007 for example who report evidence for significant lateral compositional differences around 410 km). There are other potentially important factors as well including minor phases, partial melt, fluids, and anelastic effects on seismic velocities, the latter of which may be coupled to H content. It is very difficult to untangle all these factors, and the authors' approach which just focuses on three specific factors will likely suffer from a lot of tradeoffs with these other variables. Also, the elastic properties for garnet/majorite could be different depending on what experimental studies are chosen as reference values. Different results may also be obtained depending on which seismic model is used. On balance, there are too many assumptions and tradeoffs to reliably constrain the lateral variation in wadsleyite, temperature, and hydrogen content by this approach, and hence the conclusions drawn about the nature of the transition zone are premature.

4. The paper could also use some improvement in writing style for clarity.

Reviewer #3 (Remarks to the Author):

Dear Editor,

Thank you for the opportunity to collaborate to the review process of this interesting study.

Dear Authors,

I have read your manuscript with interest, and I have enjoyed it. You present an experimental mineral physics-based quantitative interpretation of the seismic velocity heterogeneity and the topography of the 410 km discontinuity at the top of the mantle transition zone. I find both your experimental work and the modeling very well designed, and I consider the final results of your modeling a very important contribution to our understanding of the compositional heterogeneity of this special region of the Earth's interior. Your results can impact both our understanding of the current composition and structure of the mantle, and drive future global dynamic modeling of the mantle and its evolution.

As it always happens, I also have some comments and questions, and I believe your manuscript can be definitely improved to have the maximum impact on the widest audience in the geophysics and geosciences

community.

My first (and by far the most important) comment is about properly acknowledging the existing investigations of the elastic properties of wadsleyite as a function of chemical composition and pressure. I believe you should explicitly cite the existence of previous experimental studies (also high-pressure studies based on Brillouin scattering). I would suggest you to use the current reference 28 and say that the available literature is cited, and put in context (i.e. compared with your own room temperature high pressure results) in that article. You can find space for this necessary reference between line 73 and line 78 (where reference 28 is currently cited) at the cost of a little rearrangement of the text. I strongly recommend you to consider this addition; I believe that the manuscript can give (in its current form) a somewhat biased impression about the current knowledge of the elastic properties of wadsleyite (including the effects of Fe and H₂O contents). Finally, I would suggest you to briefly motivate your choice of the source data for the elastic properties of the other phases in your average rock seismic speeds calculations. Mentioning the existence of a rich literature focusing on the elastic properties of garnet-majorite at high pressure (and referring to some review articles referencing it) would be a very nice addition to your current manuscript. Here I deliberately decided not to mention any study in particular, because I consider this issue very important, above the choice of specific articles to be cited; in addition the literature is nowadays easily accessible.

I list all my other comments, suggestions and questions below.

Main text

- 1) At line 27 in the abstract you report "... 10 vol.% less wadsleyite..." which is not reported in the main text. I would expect more details in the discussion and a reference to the main results in the abstract. In this case it seems the opposite. The discussion is in general terms, and then abstract contains a numerical value which I cannot find elsewhere in the whole manuscript.
- 2) At line 30 I am not sure I understand "also stable" here. Does it mean this a stable (fit) solution compared to that for volume %?
- 3) At lines 38-39 I would substitute "... thus bridgmanite..." with "...that is bridgmanite...".
- 4) At line 59 "T" should be in italics.
- 5) At lines 77-78 "... have moderate effects on the P dependence... and shear modulus (G) at 300 K...". Please my initial general comment.
- 6) At line 79 it should be "First-principles".
- 7) At lines 89-91 I would suggest "By integrating the experimental... we mapped the global distribution of wadsleyite abundance, T anomaly...".
- 8) At lines 100-101 and at other places in the manuscript I would suggest a different notation for the partial derivatives of the elastic moduli. As an example I think $(\partial K_S / \partial P)_{T_0}$ is more clear than $\partial K_{S0} / \partial P$. This is because K_{S0} is a constant. In addition $\partial K_S / \partial T$ should be substituted with $(\partial K_S / \partial T)_{P_0}$. The same applies to the derivatives of G.
- 9) At line 105 the same comment as (8) also applies to the derivatives of the elastic coefficients C_{ij} .
- 10) At line 134 and at other places in the manuscript "T" refers to the difference (anomaly) with respect to the average temperature at 410 km depth. I find that the anomaly should be indicated by the symbol "Delta T" rather than T. Indeed, this is the symbol you use later in the manuscript (e.g. at line 292) .
- 11) At lines 135 and 137 the manuscript refers to Equations S1 to S5. Unfortunately I was not able to retrieve them in the supplementary material (file: "320023_0_other_1_qw7pp2.zip").
- 12) At line 227 I would suggest "... the subduction of Si-poor oceanic mantle lithosphere...".
- 13) At line 228 I wonder if it should be "contribute" instead of "contributed".
- 14) At line 239 I would suggest "...unpolarized IR absorption spectra...".
- 15) At lines 240-241 I would suggest "Moessbauer spectroscopy was performed on..." instead of "The Moessbauer spectrum was collected on...".
- 16) At line 255 I would substitute "allowable" with "possible".
- 17) At lines 259-260 it is not clear why you define your scattering geometry as "50 degree symmetric" if the average effective (calibrated) scattering angle is different from 50 degree well above the uncertainty (and the accuracy) on the scattering angle achievable in your system.
- 18) At lines 277 and 281 I would suggest you to modify the notation of the partial derivatives (see comment 8 above).
- 19) At lines 286-287 I would suggest "in... steps" rather than "in... step".

20)At line 290 (Equation 1) and 305 (equation 2) velocity anomalies of (modified) pyrolite due to the change of wadsleyite volume fraction are determined as weighted averages of the individual minerals velocities. The approach used here is different with respect to the commonly used volume-weighted averaging of the elastic moduli and then calculate velocities of the rock as a homogeneous effective medium. I would suggest you to explain the reason of your choice. Are the results of the two approaches very similar in this case?

21)At line 295 the reference to Tables S3 and S4 is unclear. The references are to the set of thermoelastic parameters used to determine the coefficients presented in Equation 1. The relationship between the content of the Tables and the coefficients involves analysis of a set of data from the literature; for this reason I would suggest to explicitly stating this in the manuscript. I would suggest a form similar to "... (The parameters used to determine these coefficients are reported in Table...".

22)At line 296 I would suggest substituting "Delta V_p" with "Delta V_{p_pred}".

23)At lines 307 and 308 I find the same unclear reference to the content of Tables S3 and S4 as mentioned in comment 21.

24)At lines 313-314 I would substitute ", adding water into the wadsleyite crystals is going to elevate the 410..." with ", increasing the H₂O content of wadsleyite elevates the discontinuity...".

25)At line 323 I wonder why you do not add the value of "d410_obs_error" in %, that is 1.3 %. This would give an immediate idea of the relative size of the uncertainties on the three parameters.

26)At lines 341-342 I would suggest "... and are shown in Fig. 3."

27)At line 359 I find the caption of Figure 1 unclear. I would describe this figure as "Relative variation, in percent, of wadsleyite seismic velocity as a consequence of water content and temperature anomaly at the P-T average conditions of the 410 km seismic discontinuity".

28)At line 361, I am not sure that panel a2, b2 and c2 are clearly presenting the uncertainties as stated in the caption. It seems to me that the color scales are not the correct ones. I apologize if I misunderstood something here.

Supplementary information file

29)At line 13, Figure S1, I wonder if this spectrum should be labeled with V_{s1} + V_{s2}. The full width at half maximum of the V_s peak is comparable to that of V_p. Due to the expected width scaling with frequency (or velocity) I would imagine this is a case of partial overlap of the two shear modes.

30)In Table 2, 3 and 4, the notation of the derivatives (in the columns titles) could be improved (see my comment 8, above)

31)In my opinion table 3 needs an additional note. "The entries at row 7 (reference 1) have been compared with an existing literature focusing on the pressure dependence of the elastic properties of wadsleyite at ambient temperature. The references to these studies can be found in Reference 1". This cannot be overlooked, because these results (covering a range of wadsleyite compositions, i.e. H₂O contents and Fe contents) are part of the information which ultimately affects the modeling presented in this study, which includes the effect of H₂O content and of Fe content (which is neglected also based on the whole available set of previous studies).

32)Equations S1 to S5 are missing in my version of the supplementary information file.

Response to reviewers' comments on manuscript NCOMMS-21-26303-T: "Constraining Composition and Temperature Variations in the Mantle Transition Zone" by Wen-Yi Zhou, Ming Hao, Jin S. Zhang, Bin Chen, Ruijia Wang, and Brandon Schmandt.

We thank the editor and all three reviewers for their helpful suggestions and thoughtful comments, which helped us substantially improve the manuscript. Our detailed, point-by-point responses are given below in purple color, and the changes made in the manuscript are shown in red color.

Response to reviewer 1

This manuscript presents experimental results on the single-crystal elasticity of hydrous Fe-bearing wadsleyite at high P-T conditions for the first time. These results were then compared with Vp and Vs velocities from seismic tomographic models and a model for the depth variation of the 410-km discontinuity to place constraints on the geographical distribution of wadsleyite proportion, temperature anomaly, and water content in the upper mantle transition zone (MTZ). This is a very well written manuscript with robust experimental data. The model presented is neat and powerful to resolve the non-uniqueness associated with inversions. I thus would like to recommend the publication of this manuscript in Nature Communications. Below are some comments/suggestions to consider/clarify.

We thank the reviewer for the concise summary and positive evaluation of our work.

1. *I think the manuscript needs some work to better quantify the uncertainties for the wadsleyite proportion, T anomaly, and water content. It seems to me the current uncertainties come from the parameter distributions during the grid search. However, there are other sources of uncertainties not considered in this procession. For example, the resulted temperature ($\sim 1870 \pm 150$) and water content ($\sim 0-0.6$ wt%) in the upper MTZ (Fig. 3) are much higher than the highest temperature (700 K) and water content (0.18wt%) in the experiments. This could result in some uncertainties during extrapolation and should be included in the final inversion results. In addition, the uncertainties in the seismic tomography models for absolute values of Vp/Vs and 410 km depth were not discussed. These could be another significant source of error, although they should not invalidate the main distribution feature obtained from the study.*

We thank the reviewer for pointing out the issue of uncertainty. We considered the uncertainties originated from seismic observations in our models, details are shown in the Methods section (Line 380). In terms of the Fe and water effects on the Vp and Vs of wadsleyite, as shown in Zhou et al. (2021)¹, the average difference between model-predicted values and the experimental data from previous studies is 0.7%. In terms of the T (temperature) effect, as shown in Fig. 1 in the manuscript, the average difference between the model-predicted values and the experimental data is even less and on the level of 0.1%. However, given the limited T range we experimentally explored as pointed out by the reviewer, in the parameter space used in this study (ΔT from -150K to 150 K compared with 1870 K, water content from 0 to 1 wt%), the calculated Vp and Vs uncertainties of our wadsleyite sample at ~ 1870 K, 14 GPa are higher than 0.1%, and on the level of $\sim 1\%$ without considering the composition effect-induced uncertainties. Considering the experimental uncertainties of the raw experimental data in previous studies, to address the issue raised by the reviewer, as a conservative estimate, we

assigned an assumed 1 standard deviation of 10% for each of the parameters (a, b, c, d, A, B, C, D, e, and f in equation 1-5) used in our modeling work, and then took the following procedure to evaluate the possible influence of these uncertainties on the robustness of our calculation:

1) Instead of fixing the a, b, c, d, A, B, C, D, e, and f parameters to the best-fit values, we randomly sampled the value of each parameter from a normal (Gaussian) distribution based on the best-fit value and the assumed 10% standard deviation.

2) We then re-calculated the global maps of wadsleyite proportion, ΔT (temperature anomaly), and water content in the MTZ based on the randomly selected set of values in step 1).

3) We repeated steps 1) and 2) a few times, and found that they yielded nearly identical results compared to those shown in the manuscript, which suggests that these uncertainties are unlikely to affect the robustness of this study. The simple explanation is that the influence of the global variance in seismic structure is large compared to the uncertainties in the experimentally derived model parameters. Two typical test results are shown below:

Table 1: 2 different sets of parameters used in the tests.

Parameters	a	b	c	d	A	B	C	D	e	f
Best-fit	-0.0027	-0.0028	0.0397	0.9750	-0.0040	-0.0045	0.0868	0.4850	0.107	20
Test 1	-0.0026	-0.0029	0.0358	0.9439	-0.0036	-0.0054	0.0957	0.4266	0.102	19
difference	-4%	5%	-10%	-3%	-11%	18%	10%	-12%	-4%	-5%
Test 2	-0.0028	-0.0027	0.0341	1.0444	-0.0036	-0.0049	0.0741	0.5405	0.122	25
difference	3%	-3%	-14%	7%	-11%	9%	-15%	11%	14%	25%

Fig. 1. Global wadsleyite proportion, ΔT , and water content maps of the upper MTZ based on the mineral physics parameters shown in Table 1.

These additional tests we conducted above were included in the Supplementary Information (Supplementary Discussion 1, Supplementary Fig. 9, Supplementary Table 7), and mentioned in the Methods section (Lines 380-383).

2. *Better add error bars or state the uncertainties for the moduli and pressures in Fig. 1, Fig. S2, and Fig. S3.*

We thank the reviewer for pointing out this. We updated the figures with error bars in Fig. 1 and Supplementary Fig. 3. Supplementary Fig. 2 is not revised because the error bars in Supplementary Fig. 2 are significantly smaller than the symbols. We also revised all the figure captions by adding this sentence: "Error bars smaller than the symbols are not shown."

3. *Fig. S1: It looks that there are two peaks for the V_p signal on the left hand side of the spectrum and two overlapping peaks for V_p on the right hand side. What is the likely cause for the peak splitting? How V_p is obtained based on a spectrum like this?*

The spectrum shown in Supplementary Fig. 1 has been processed under low pass filter, which results in the artificial splitting shown in the original Supplementary Fig. 1. During data processing, we conducted peak fitting to obtain 1 V_p with 1 or 2 V_s in each spectrum. We added more spectra in the updated Supplementary Fig. 1 in the revised manuscript. The figure caption has been revised to "Representative Brillouin spectra of single-crystal wadsleyite at different high P-T conditions. The spectra have been processed with a low pass filter. The collection time is approximately 20 minutes for each spectrum."

4. *Line 112: It should be "adding 1 wt% water in wadsleyite" instead of "adding 1 wt% in wadsleyite".*

Thanks. We have corrected the sentence in the updated manuscript (Line 113).

5. *Line 134: It should be "T anomaly" instead of "T".*

Thank you. We have corrected it in the updated manuscript (Line 137).

6. *Figure 3a2 and Figure S4a2: I believe the color bar is incorrect for these two figures. The uncertainty should not be as large as ~55-65%. I suspect a 53% should be subtracted.*

We apologize for this mistake and have corrected the color bar in Fig. 3 and Supplementary Fig. 4.

Response to reviewer 2:

This paper reports measurements of the single-crystal elastic properties of an H-bearing wadsleyite at high pressures and temperatures. The results are used to develop a model for the variation of wadsleyite fraction, H content, and temperature in the Earth's upper transition zone. The work builds on an earlier study by Zhou et al 2021 which reported measurements on the same samples at ambient temperature and high pressure (although this should be stated more clearly in the present paper). The elasticity data appears to be of generally high quality and the

measurements on hydrous wadsleyite at simultaneous high P-T conditions have not been reported before and this certainly warrants publication. The authors also apply their data in an attempt to simultaneously constrain wadsleyite content, hydrogen content, and temperature in the upper part of the transition zone by combing their data with elasticity data for garnet/majorite (the other major phase in the region) and comparing with seismic velocities at 450 km depth as well as the depth of the 410-km discontinuity. I find this part of the paper to be more speculative and now well constrained. As a result, I don't think the paper warrants publication in Nature Communications.

We thank the reviewer for the concise summary of our work. We agree with the reviewer that the fact that the wadsleyite samples used in this work are same as the samples used in Zhou et al. (2021) should be stated more clearly in the present manuscript. To clarify this, we added a sentence at the beginning of the Methods section (Lines 252-253) and added a sentence in the data analysis (Lines 307-308). We also specifically mentioned that part of the sample characterization experiments (EPMA, Mossbauer experiments, and part of the FTIR experiments) have been published in Zhou et al. (2021)¹ (Lines 264-266). To thoroughly address each specific issue raised by the reviewer, we have not only conducted additional experiments, but also performed extra modeling tests as well as substantially revised the text of the manuscript by acknowledging the technical limitations and uncertainties. We feel the manuscript has been much improved and appreciate the reviewer's comments and suggestions.

Major comments:

1. Characterizing the water content in the sample is crucial for this kind of work, and the authors mainly refer to their earlier paper (Zhou et al EPSL 21) for details of the water content determination. However, I don't feel that characterization in that paper is sufficient, and more details are needed here. Determination of water content by FTIR requires specific calibration which can vary significantly between different laboratories. The authors here apparently only use 1 calibration. Why was that calibration chosen and how much do the results change when using a different calibration? If the presence of Fe³⁺ complicates the calibration by IR, then another technique such as SIMS should be used to determine the H₂O content.

We thank the reviewer for pointing out the importance for accurate determination of the water content in the samples used in this study. The calibration method (Libowitzky and Rossman, 1997)² used in Zhou et al. (2021)¹ is the same as the ones used in previous sound velocity studies on hydrous wadsleyite (Buchen et al., 2018³ and Mao et al., 2008a⁴). The water content of the sample used in Buchen et al. (2018)³ is 0.23 wt%, and the water content in sample WH833 in Mao et al. (2008a)⁴ is 0.37 wt%, both are close to the water content of the sample used in this study. To address the reviewer's concern with respect to the use of different calibration method, we also tried another calibration method outlined by Deon et al. (2010)⁵, which was used in Chang et al. (2015)⁶. The use of calibration by Deon et al. (2010)⁵ yields similar water content as shown in Table 3 in this response letter, which is Supplementary Table 5 in the Supplementary Information. We added this information in the revised manuscript (Lines 263-264).

The reviewer is correct that, if the SIMS has been carefully calibrated to a wadsleyite sample with known water content, it would give more precise results compared with FTIR

measurements calibrated by glasses with known water content. However, as pointed by Chang et al. (2015)⁶ in section 3.1: “The discrepancy between the water content determined by various FTIR calibrations and by SIMS appears negligible when the water content is lower than ~0.5 wt%”. Therefore, such improvement in terms of precision, only becomes significant when the water content in wadsleyite exceeds 0.5%. As shown in Table 4 in Chang et al. (2015)⁶, which is also shown in Table 2 below, the difference between SIMS and FTIR results is less than 0.01 wt% for the same hydrous wadsleyite samples with water content less than ~0.5 wt%. Therefore, we do not expect dramatically different results from SIMS measurements compared with the FTIR determined value of 0.15 (4) wt% in this study.

Sample	Water content (FTIR)	Water content (SIMS)
WS3056	0.005	0.006(3)
WH833	0.37(1)	0.38(8)

Table 2. Comparison of the water contents for the same wadsleyite samples determined by polarized FTIR (Jacobsen et al., 2005)⁷ and SIMS (Chang et al., 2015)⁶, respectively.

Quantification of water in anisotropic minerals by IR spectroscopy requires consideration of orientation effects, which apparently was not done here as only unpolarized spectra were used.

Quantification of water in anisotropic minerals using FTIR indeed requires consideration of orientation effects. Therefore, in order to obtain reliable water content from unpolarized FTIR spectra, we conducted unpolarized FTIR measurements on 9 randomly oriented double-side polished wadsleyite crystals in Zhou et al. (2021)¹. As suggested by Smyth et al. (2014)⁸, unpolarized FTIR measurements on 9-10 randomly oriented wadsleyite crystals yield very similar water content as what was obtained from polarized FTIR measurements (See Fig. 2 in Smyth et al. (2014)⁸, and Fig. 2 below). The consistency between polarized FTIR and unpolarized FTIR measurements was supported by Buchen et al. (2017)⁹ as well. Similar water contents were obtained from unpolarized FTIR (0.25 (4) wt%) and polarized FTIR (0.24 (2) %) measurements for the same wadsleyite sample used in Buchen et al. (2017)⁹.

Fig. 2. FTIR spectra of sample 1039A measured using polarized FTIR (a) and unpolarized FTIR (b) on 9-10 randomly oriented grains (modified after Smyth et al. (2014)⁸). The average of polarized and unpolarized spectra are shown as red curves. They are very similar when enough grains (9-10) are measured.

To address the reviewer’s concern and better characterize water content of our wadsleyite sample, we polished another 8 randomly selected wadsleyite single crystals for additional unpolarized FTIR measurements. The updated water content of the wadsleyite sample used in this study is 0.15 (4)% using a total of 17 crystals, about 0.01 wt% or 100 ppm higher than the water content determined in Zhou et al. (2021)¹ from 9 crystals (0.14 (4) %). This small change is within the water content uncertainty of 400 ppm. To test if this small change would affect the composition dependent elasticity model of wadsleyite, which is presented in Zhou et al. (2021)¹ and used in this study, we repeated the modeling work conducted in Zhou et al. (2021)¹, and found that the results remain the same. These new measurements (Fig. 3 and Table 3 below) have been added into the Supplementary Information as Supplementary Fig. 5 and Supplementary Table 5. The manuscript has also been revised accordingly (Lines 266-275).

Fig. 3. Unpolarized spectra of the 17 randomly oriented wadsleyite platelets. The water content shown in the legend were calculated using the calibration procedure outlined in Libowitzky and Rossman (1997)².

Sample Name	Water content (wt %)	
	Libowitzky and Rossman (1997) ² calibration	Deon et al. (2010) ⁵ calibration
Wad25	0.19	0.17
Wad19	0.13	0.13
Wad35	0.10	0.11

Wad31	0.13	0.11
Wad48	0.16	0.14
Wad 34	0.18	0.15
Wad24	0.16	0.15
Wad37	0.18	0.16
Wad1	0.22 ¹	0.12
Wad7	0.09 ¹	0.08
Wad8	0.08 ¹	0.06
Wad9	0.11 ¹	0.10
Wad10	0.15 ¹	0.12
Wad11	0.16 ¹	0.12
Wad14	0.18 ¹	0.17
Wad17	0.12 ¹	0.11
Wad15	0.15 ¹	0.14
Average	0.15 (4)	0.13 (3)

Table 3. Water contents of 17 different wadsleyite crystals determined using unpolarized FTIR with different calibration methods.

Finally, it should be noted that the speciation of the H-defect location could affect the measured properties. What can be said about this from the IR spectra? This is all important b/c without a very good calibration and some knowledge of H speciation, it is difficult to compare the present results with those of other studies and the literature becomes confused with incompletely characterized samples.

We agree with the reviewer that the H-defect location can be important. For example, different high-P sound velocities have been reported for the ringwoodite samples with very similar water and Fe contents (Mao et al., 2012¹⁰ and Schulze et al., 2018¹¹) using the same experimental method (Brillouin spectroscopy). The most plausible explanation is the different hydration mechanisms for the samples used in these 2 different studies (Schulze et al., 2018)¹¹. As a result, for ringwoodite, the simple composition dependent elasticity model which ignores different H-defect locations, similar to the one used in Zhou et al. (2021)¹, cannot fit both the data in Mao et al. (2012)¹⁰ and the data in Schulze et al. (2018)¹¹ at the same time. However, this is not the case for wadsleyite. Our composition (Fe and water) dependent Vp and Vs model presented in Zhou et al. (2021)¹ could fit all previously measured Vp and Vs data very well (Sawamoto et al., 1984¹²; Zha et al., 1997¹³; Mao et al., 2008a⁴, 2008b¹⁴; Mao et al., 2011¹⁵; Wang et al., 2014¹⁶; Buchen et al., 2018³; Zhou et al., 2021¹) within a wide range of Fe (Fe#=0-11.2) and water content (0-2.9 wt%). At least, for all the hydrous wadsleyite samples that have been measured in the past few decades, the hydration effect on the Vp and Vs of wadsleyite can be described

by the simple model used in Zhou et al. (2021)¹, which ignored the effect from different H-defect locations. Therefore, for wadsleyite, unless new experimental evidence shows up, there is no need to involve additional parameters to separately describe the effects from different H-defect locations on the Vp and Vs.

2. According to Figure 1, it looks to my eye that there is a bigger reduction in moduli and velocity from 500-700 K than from 300-500K. This is somewhat surprising especially as the authors assume a linear variation with temperature. Is that correct? A separate issue is whether there could be some change to the sample on heating, for example change in the amount or speciation of hydrogen. What was done to verify that the sample did not change significantly upon heating?

According to the fitting results shown as solid lines in Fig. 1 in the manuscript, the reduction in K (or G) as well as the Vp (or Vs) from 500K to 700K is similar to the reduction from 300-500 K. The different impression that the reviewer had could be caused by 400K isotherm (blue line).

We found negligible water content change in the samples after the high P-T experiment. We provided the following three additional experimental evidence below, which has been added to the Methods section (Lines 315-332) and Supplementary Information (Supplementary Figs. 6-8) as well.

1) We collected FTIR spectra on Wad34 at 300 K before heating, and at 300 K after it was heated to 400K, 500K, and 700K. The water content was determined to be 1800 ppm before heating, and remains the same after multiple heating and cooling cycles. 1800 ppm is within the uncertainty of the 1500 (400) ppm water content measured from unpolarized FTIR measurements of 17 crystals. The spectra are shown in Fig. 4.

Fig.4. FTIR spectra of the wadsleyite crystal Wad34 after several heating and cooling cycles. Black line: FTIR spectrum collected before heating; Blue line: FTIR spectrum collected after the

sample had been heated at 400 K and then cooled down to 300 K. Yellow line: FTIR spectrum collected after the sample had been heated at 500 K and then cooled down to 300 K. Red line: FTIR spectrum collected after the sample had been heated at 700 K and then cooled down to 300 K.

2) After we completed the 700 K 9.4 GPa Brillouin spectroscopy experiments of Wad6 (-0.3420, 0.8718, 0.3506) and cooled it down to 300 K, we collected 5 additional Brillouin spectra of sample Wad6 at 9 GPa 300 K along 5 additional different crystallographic directions. Due to the short collection time, the spectrum quality only allows reliable determination of Vs at these 5 different directions. The obtained Vs values (red squares in Fig. 5) nicely matched the C_{ij} model-predicted Vs values at 9 GPa 300 K (black and red lines in Fig. 5), as well as the Vs at 9.5 GPa 300 K measured before the sample was heated (black triangles in Fig. 5). Thus, the water content is unlikely to be significantly changed during the high P-T experiments.

Fig. 5. Change of velocities as a function of laboratory Chi angle. Solid lines: velocities predicted by the best-fit C_{ij} model at 300 K and 9GPa from ambient-T high-P experiments in Zhou et al. (2021) (before high P-T experiments in this study); Triangles: velocities measured at 300 K and 9.5 (2) GPa in Zhou et al. (2021) (before high P-T experiments in this study); Red squares: velocities measured at 300 K and 9 GPa after the sample has been heated to 700 K 9.4 GPa in this study.

3) We conducted additional Brillouin spectroscopy measurements on sample Wad9 (0.0748 0.3154 -0.9460) at 400 K and 0 GPa. This is the only crystal survived after the entire series of high P-T experiment. This newly collected set of data at 400 K and 0 GPa was not utilized in deriving the high P-T C_{ij} model of wadsleyite as presented in the manuscript and shown in Supplementary Fig. 3. If significant water loss has taken place, this set of velocities would be faster than the C_{ij} model-predicted Vp and Vs values shown as the black lines in Fig. 6 below. Clearly, it is not the case. We originally planned to conduct FTIR measurements on this crystal after it was recovered at room P-T condition, unfortunately it was lost during the transportation from the DAC to the CaF₂ glass plate for FTIR measurements.

Fig 6. Change of velocities for wadsleyite sample Wad9 (0.0748 0.3154 -0.9460) as a function of laboratory measurement Chi angle at 400 K and 0 GPa. Solid lines: the velocities predicted by the best-fit C_{ij} model at 400 K and 0 GPa based on high P-T experiments in this study; Squares: velocities collected at 400 K and 0 GPa after the completion of the high P-T experiments.

3. The authors attempt to constrain the wadsleyite fraction, water content, and temperature from seismic velocity data at 450 km depth and the depth of the 410-km discontinuity. In doing so, several assumptions or simplifications must be made. A major one is the assumption that chemical composition (Fe, Al, Ca, Na, etc) does not change laterally. This is unlikely to be true (see Schmerr and Garnero, 2007 for example who report evidence for significant lateral compositional differences around 410 km). There are other potentially important factors as well including minor phases, partial melt, fluids, and anelastic effects on seismic velocities, the latter of which may be coupled to H content. It is very difficult to untangle all these factors, and the authors' approach which just focuses on three specific factors will likely suffer from a lot of tradeoffs with these other variables. Also, the elastic properties for garnet/majorite could be different depending on what experimental studies are chosen as reference values. Different results may also be obtained depending on which seismic model is used. On balance, there are too many assumptions and tradeoffs to reliably constrain the lateral variation in wadsleyite, temperature, and hydrogen content by this approach, and hence the conclusions drawn about the nature of the transition zone are premature.

We are glad to know that the reviewer holds the same view as us on the chemically heterogeneous nature of the Earth. There are 4 main points raised by the reviewer in this paragraph, and we now address each point in the following sections:

(1) The reviewer pointed out that we did not take the compositional variations in terms of Fe, Al, Ca, Na, etc. into account. This is not true. The major difference between this and previous studies (e.g. Wang et al., 2021¹⁷) is that we considered the possible petrological deviation from the idealized pyrolite upper mantle model. Similar to what the reviewer thinks, we believe that the Earth is far more complicated than what can be described by a simplified pyrolite model. In this study, at the upper MTZ depth ranges we looked at, the petrological deviation from the idealized pyrolite model can be manifested as the lateral change of the wadsleyite/olivine content. Numerous high P-T phase equilibrium experiments on pyrolite, harzburgite and mid-

ocean ridge basalt have revealed that the upper MTZ can be viewed as a simplified mineralogical binary mixing system between wadsleyite and majoritic garnet (e.g., Ishii et al. 2018¹⁸, 2019¹⁹). The lateral variations of wadsleyite and majoritic garnet content are directly linked to the change of major elements concentrations, including Fe, Al, Ca, etc. For example, more wadsleyite and less majoritic garnet would suggest less Na, Al, Fe, Si, and Ca, yet more Mg. It is also worth noting that, the compositions of wadsleyite and majoritic garnet are not assumed as end member compositions in this study, they are adopted from the most recent phase equilibrium experiments (Ishii et al. 2018¹⁸, 2019¹⁹). Therefore, the compositional variations have been taken into account through the petrological/mineralogical variations in terms of wadsleyite content in this study. We added this information in the manuscript (Lines 120-124).

(2) The reviewer pointed out that minor mineral phases, partial melt, fluids, and anelastic effects should be considered in the models. We are fully aware of the possible existence of the minor mineral phases (e.g. alphabetical phases in the slabs), partial melt and fluids in the mantle, in particular on a local scale. The anelastic effects such as grain boundary sliding (Faul and Jackson, 2005²⁰; Karato, 2012²¹) can be important for localized regions with very high Ts (plume) and relatively small mineral grain sizes (strongly sheared regions near fast subduction zones). However, the seismic velocity Vp and Vs models used in this study by Koelemeijer et al. (2016)²² are degree 12 models, and they are only sensitive to the composition or T anomalies on the scale of several hundred to several thousand kilometers. Small-scale velocity anomalies caused by the existence of minor minerals/fluid phases/anelastic effects can be important, but are unlikely to strongly affect the large-scale anomalies we focused on in this study. Therefore, we did not consider these effects in this study. However, we agree with the reviewer that, in the future, if researchers were going to use a similar approach to study the compositional and T variations in the mantle on a regional scale with significantly improved spatial resolution, these effects probably need to be taken into consideration.

(3) The thermoelastic parameters of garnet with different compositions are listed in Supplementary Table 4. In terms of our choice of the source data for calculating the sound velocities of Fe-Ca bearing majoritic garnet in the upper MTZ, our preference was given to the most recent direct sound velocity measurements at simultaneously high P-T conditions. For end member majorite and pyrope, their ambient-condition K_{S0} and G_0 are calculated using the elasticity data of majorite-pyrope solid solutions: $Maj_{38}Py_{62}$ (Liu et al., 2000)²³, $Maj_{50}Py_{50}$ (Sinogeikin and Bass, 2002p²², 2002t²³), $En_{80}Py_{20}$ (Sinogeikin and Bass, 2002t)²⁴. Their P and T derivatives of K_S and G are based on the sound velocity data collected on a pyrope garnet ($Maj_{50}Py_{18}Gro_{25}Alm_6$) by Irifune et al. (2008)²⁵ up to ~18 GPa and 1673 K, which are close to the P-T conditions we explored in this study. As suggested by Liu et al. (2019)²⁶, both the sound velocities and elastic moduli of the garnet within the majorite-pyrope solid solution increase almost linearly with Al content within analytical uncertainties. The thermoelastic properties of the almandine end member are adopted from the most recent high P-T sound velocity measurements up to 19 GPa and 1700 K by Arimoto et al. (2015)²⁷. For the end member grossular, we used the values presented in the most recent high P-T elasticity measurements up to 10 GPa 1000K by Gwanmesia et al. (2014)²⁸. We did not use the elasticity data reported in Zhou C.Y. et al. (2021)²⁹ for end member $MgSiO_3$ majorite due to the following reasons: 1.

Although the sound velocity measurements were conducted up to 19 GPa and 2000 K, their reported Vp and Vs values from 900 K to 2000 K cannot be successfully fitted by finite strain EOS. 2. The Vs of majorite at 2000K cannot be fitted by a second order Tylor expansion of K and G either. Zhou C.Y. et al. (2021)²⁹ has acknowledged these issues in their data, and attributed them to the unresolved inaccuracies in high-T measurements or the possibly much greater effect of T on G, which needs future investigation. Therefore, we did not use the result presented in the most recent high P-T elasticity study for end member majorite from Zhou C.Y. (2021)²⁹. We have included the motivation for choosing the parameters for garnet in Supplementary Table 4.

Overall, garnet is a very well-studied (probably the most intensively studied) mantle mineral due to its high symmetry and its geological importance. Its elastic properties at high P-T conditions are highly consistent across different studies. Using the thermoelastic parameters determined from different studies has very small influence on the calculated velocities of garnet. For example, as shown in Table 4 below, if we adopt the P and T derivatives of Ks and G for pyrope and majorite from Liu et al. (2015)³⁰ instead of Irifune et al. (2008)³¹, as well as the thermoelastic parameters from Kono et al. (2010)³² for grossular instead of Gwanmesia et al. (2014)²⁸, the Vp and Vs of the majoritic garnet (Maj₅₅Py₁₄Gro₁₈Alm₁₃)¹⁸ at 15.2 GPa and 1870 K only change by 0.4% and 0.8 %, respectively, and are within the experimental uncertainties.

Velocities of Maj ₅₅ Py ₁₄ Gro ₁₈ Alm ₁₃ at 15.2GPa 1870K	Based on Supplementary Table 4	Based on Liu et al. (2015) ³⁰ and Kono et al. (2010) ³²	difference
Vp	9.32 km/s	9.36 km/s	0.4%
Vs	4.89 km/s	4.93 km/s	0.8%

Table 4. Comparison of sound velocities of the majoritic garnet in pyrolite (Maj₅₅Py₁₄Gro₁₈Alm₁₃) at 15.2 GPa 1870 K using different thermoelastic parameters.

(4) The reviewer suggested that our results may change with the use of different seismic models, which we agree and have explained our choice of seismic models in the manuscript (Lines 139-143). The Vp and Vs tomography models by Koelemeijer et al. (2016)²² used inversion framework of the S40RTS Vs tomography model. The most recent global MTZ discontinuity topography model by Huang et al. (2019)³³ also used S40RTS tomography model (Ritsema et al., 2011)³⁴ to correct for the lateral seismic heterogeneities in the upper mantle. The three seismic models used in this study are internally consistent with each other, which is crucial for this study. We are not aware of any other combinations of seismic models that are internally consistent and provide all the parameters needed for this study.

To summarize, we agree with the reviewer that it would be ideal if we can separate the compositional effects caused by each geologically important element in the Earth's interior, it would also be great if we can take all the minor phases as well as anelastic effect into account, it would be even nicer if we can use the Vp Vs tomography and the 410 topography models with the highest spatial resolution possible, maybe down to 50 km or less. However, these are impossible given the limited seismic observations we can use, as well as the limited mineral physics constraints we have from experiments. With these limitations, we have to carefully

select internally consistent seismic models with sacrifice of the spatial resolution, we also have to identify the first-order controlling factors for these long-wavelength large scale seismic observations from the mineral physics perspective and neglect secondary factors such as the existence of minor phases. The results presented in this study are preliminary, and by no means can we consider it resembles all the details of the upper MTZ. However, this study is our first step toward a multi-dimensional compositional model inside the Earth. Mineral physics and seismologists will need to work together toward this ultimate goal in the future, so we added some sentences to highlight the limitations of this study (Lines 243-249) in the manuscript. We again thank the reviewer for raising these concerns and hopefully our response has adequately addressed all of them.

4. The paper could also use some improvement in writing style for clarity.

To improve the clarity and readability of the paper, during the revision, we have iterated the manuscripts between all coauthors multiple times. We feel the current manuscript has been greatly improved.

Response to reviewer 3:

Dear Authors, I have read your manuscript with interest, and I have enjoyed it. You present an experimental mineral physics-based quantitative interpretation of the seismic velocity heterogeneity and the topography of the 410 km discontinuity at the top of the mantle transition zone. I find both your experimental work and the modeling very well designed, and I consider the final results of your modeling a very important contribution to our understanding of the compositional heterogeneity of this special region of the Earth's interior. Your results can impact both our understanding of the current composition and structure of the mantle, and drive future global dynamic modeling of the mantle and its evolution.

As it always happens, I also have some comments and questions, and I believe your manuscript can be definitely improved to have the maximum impact on the widest audience in the geophysics and geosciences community.

We thank the reviewer for the concise summary, positive evaluation, and helpful suggestions.

My first (and by far the most important) comment is about properly acknowledging the existing investigations of the elastic properties of wadsleyite as a function of chemical composition and pressure. I believe you should explicitly cite the existence of previous experimental studies (also high-pressure studies based on Brillouin scattering). I would suggest you to use the current reference 28 and say that the available literature is cited, and put in context (i.e. compared with your own room temperature high pressure results) in that article. You can find space for this necessary reference between line 73 and line 78 (where reference 28 is currently cited) at the cost of a little rearrangement of the text. I strongly recommend you to consider this addition; I believe that the manuscript can give (in its current form) a somewhat biased impression about the current knowledge of the elastic properties of wadsleyite (including the effects of Fe and H₂O contents).

We absolutely agree with your suggestion. We have made substantial changes in the updated Introduction section (Lines 69-73 and Line 79) in the manuscript accordingly, and hopefully we have properly addressed your concern.

Finally, I would suggest you to briefly motivate your choice of the source data for the elastic properties of the other phases in your average rock seismic speeds calculations. Mentioning the existence of a rich literature focusing on the elastic properties of garnet-majorite at high pressure (and referring to some review articles referencing it) would be a very nice addition to your current manuscript. Here I deliberately decided not to mention any study in particular, because I consider this issue very important, above the choice of specific articles to be cited; in addition the literature is nowadays easily accessible.

Thank you very much for pointing out this issue. We now added some sentences mentioning the existence of a rich literature on the elastic properties of garnet covering different compositions in the manuscript (Lines 64-69). The elastic properties of garnet at high P-T conditions are highly consistent across different studies. Detailed explanations can be found in our response to the second reviewer's comment in page 11-12 under section (3). We added a paragraph in Supplementary Table 4 to explain our choice of the elasticity data of garnet listed in Supplementary Table 4. Our preference was given to the most recent direct sound velocity measurements on garnet at simultaneously high P-T conditions.

I list all my other comments, suggestions and questions below.

Main text

1)At line 27 in the abstract you report "... 10 vol.% less wadsleyite..." which is not reported in the main text. I would expect more details in the discussion and a reference to the main results in the abstract. In this case it seems the opposite. The discussion is in general terms, and then abstract contains a numerical value which I cannot find elsewhere in the whole manuscript.

Thanks for pointing this out. We now add more details in the discussion (Lines 180-183). We also modified the abstract as suggested by the reviewer (Lines 19-24).

2)At line 30 I am not sure I understand "also stable" here. Does it mean this a stable (fit) solution compared to that for volume %?

The sentence "The geographic distribution of variations in water content is also stable" means that the water enriched regions shown in the averaged best-fit models with misfit<1 remain water enriched in the averaged best-fit models with misfit<1.5, although the absolute water content is slightly different. We have deleted this sentence in the abstract to avoid confusion and to match the length limit of the abstract.

3)At lines 38-39 I would substitute "... thus bridgmanite..." with "...that is bridgmanite..."

We revised the sentence accordingly (Line 29).

4)At line 59 "T" should be in italics.

All "T"s have been reformatted in italics throughout the revised manuscript.

5)At lines 77-78 “... have moderate effects on the P dependence... and shear modulus (G) at 300 K...”. Please see my initial general comment.

We added “ref. 29 and references cited in ref. 29” in the updated manuscript (Line 79)

6)At line 79 it should be “First-principles”.

Yes, we have corrected it accordingly throughout the revised manuscript.

7)At lines 89-91 I would suggest “By integrating the experimental... we mapped the global distribution of wadsleyite abundance, T anomaly...”.

We agree with the reviewer and have revised it accordingly (Lines 90-93).

8)At lines 100-101 and at other places in the manuscript I would suggest a different notation for the partial derivatives of the elastic moduli. As an example I think $(\partial K_S / \partial P)_{T_0}$ is more clear than $\partial K_{S0} / \partial P$. This is because K_{S0} is a constant. In addition $\partial K_S / \partial T$ should be substituted with $(\partial K_S / \partial T)_{P_0}$. The same applies to the derivatives of G.

We agree and have used this new notation throughout the manuscript and Supplementary Information.

9)At line 105 the same comment as (8) also applies to the derivatives of the elastic coefficients C_{ij} .

We have used the new notation in the revised manuscript and Supplementary Information.

10)At line 134 and at other places in the manuscript “T” refers to the difference (anomaly) with respect to the average temperature at 410 km depth. I find that the anomaly should be indicated by the symbol “ ΔT ” rather than T. Indeed, this is the symbol you use later in the manuscript (e.g. at line 292) .

We agree with the reviewer that using ΔT (temperature anomaly) is more accurate. We have modified T to ΔT in the manuscript wherever the latter is more suitable.

11)At lines 135 and 137 the manuscript refers to Equations S1 to S5. Unfortunately I was not able to retrieve them in the supplementary material (file: “320023_0_other_1_qw7pp2.zip”).

We are sorry for the typo. We have modified “Equation S1-S4” to “Equations 1-4 in Methods” (Line 147) and modified “Equation S5” to “Equation 5 in Methods” (Line 150).

12)At line 227 I would suggest “... the subduction of Si-poor oceanic mantle lithosphere...”.

We have modified the sentence accordingly (Line 241).

13)At line 228 I wonder if it should be “contribute” instead of “contributed”.

We agree with the reviewer and have changed “contributed” to “contribute” in the updated manuscript (Line 242).

14)At line 239 I would suggest “...unpolarized IR absorption spectra...”.

Thanks for the suggestion. We have revised it accordingly in the updated manuscript (Line 269-270).

15)At lines 240-241 I would suggest “Mossbauer spectroscopy was performed on...” instead of “The Mossbauer spectrum was collected on...”.

We have revised the sentence accordingly in the updated manuscript (Lines 258-259).

16)At line 255 I would substitute “allowable” with “possible”.

We substituted “allowable” with “possible” in the updated manuscript (Line 286).

17)At lines 259-260 it is not clear why you define your scattering geometry as “50 degree symmetric” if the average effective (calibrated) scattering angle is different from 50 degree well above the uncertainty (and the accuracy) on the scattering angle achievable in your system.

The Brillouin spectroscopy system was geometrically designed and aligned with an assumed ideal scattering angle of 50°, therefore it is named as “50° symmetric scattering geometry”. However, due to the limited accuracy that we can achieve with the mechanical rotational stage and optical holder, it is extremely difficult to align the Brillouin spectroscopy system with exactly 50° scattering angle. Therefore, we need to calibrate the actual scattering angle of the Brillouin spectroscopy system using a standard silica glass with known V_p and V_s . If the system is well-aligned, the difference between the calibrated scattering angle and the ideal targeted scattering angle (50°) would be less than 1°. Therefore, we name the scattering geometry as “50° symmetric scattering geometry” although the actual scattering angle is not 50° (but close).

18)At lines 277 and 281 I would suggest you to modify the notation of the partial derivatives (see comment 8 above).

Sure, we have modified all notations as suggested.

19)At lines 286-287 I would suggest “in... steps” rather than “in... step”.

We have revised the manuscript accordingly (Lines 338-339).

20)At line 290 (Equation 1) and 305 (equation 2) velocity anomalies of (modified) pyrolite due to the change of wadsleyite volume fraction are determined as weighted averages of the individual minerals velocities. The approach used here is different with respect to the commonly used volume-weighted averaging of the elastic moduli and then calculate velocities of the rock as a homogeneous effective medium. I would suggest you to explain the reason of your choice. Are the results of the two approaches very similar in this case?

We thank the reviewer for bringing up this point. We have tested both approaches and found that the velocities calculated using these two approaches are very similar. This is due to the fact that the velocities of wadsleyite and garnet are not too far away from each other under the MTZ P-T conditions. The Table 5 below shows an example of the pyrolite sound velocities calculated using the 2 different averaging approaches:

Pyrolite at 15.2 GPa and 1870K	Velocities by volume-weighted elastic moduli	Direct volume-weighted velocities	Difference
V_p	9.521 km/s	9.527 km/s	6m/s, 0.1%
V_s	5.147 km/s	5.153 km/s	6m/s, 0.1%

Table 5. Velocities of pyrolite at 15.2 GPa and 1870 K calculated using different methods.

The main reason for using the volume-weighted velocities approach is that we can avoid additional parameters that are needed for describing the T and composition dependent density. This way, we can reduce the number of mineral physics parameters involved in the modeling. We have added this important information in the manuscript (Lines 361-365) and added this Table 5 as Supplementary Table 6.

21)At line 295 the reference to Tables S3 and S4 is unclear. The references are to the set of thermoelastic parameters used to determine the coefficients presented in Equation 1. The relationship between the content of the Tables and the coefficients involves analysis of a set of data from the literature; for this reason I would suggest to explicitly stating this in the manuscript. I would suggest a form similar to "... (The parameters used to determine these coefficients are reported in Table...".

Thanks for the suggestion. We have revised the manuscript accordingly (Lines 352-354).

22)At line 296 I would suggest substituting "Delta V_p" with "Delta V_{p_pred}".

Thanks for the suggestion. We have revised the manuscript accordingly (Line 346-347).

23)At lines 307 and 308 I find the same unclear reference to the content of Tables S3 and S4 as mentioned in comment 21.

We have revised the manuscript accordingly by adding a sentence "The parameters used to determine these coefficients (A, B, C, D) are reported in Supplementary Tables 2, 4" (Lines 352-354).

24)At lines 313-314 I would substitute ", adding water into the wadsleyite crystals is going to elevate the 410..." with ", increasing the H₂O content of wadsleyite elevates the discontinuity...".

We have revised the manuscript accordingly (Lines 370-371).

25)At line 323 I wonder why you do not add the value of "d410_obs_error" in %, that is 1.3 %. This would give an immediate idea of the relative size of the uncertainties on the three parameters.

The depth of 410 is represented in km and the uncertainty is 5.4 km, we added 1.3% in the manuscript as suggested by the reviewer (Line 380).

26)At lines 341-342 I would suggest "... and are shown in Fig. 3."

Thanks for pointing out this mistake, we have added "are" before "shown" (Line 402).

27)At line 359 I find the caption of Figure 2 unclear. I would describe this figure as "Relative variation, in percent, of wadsleyite seismic velocity as a consequence of water content and temperature anomaly at the P-T average conditions of the 410 km seismic discontinuity".

We thank the reviewer for the excellent suggestion, and have revised it accordingly in Figure 2.

28)At line 361, I am not sure that panel a2, b2 and c2 are clearly presenting the uncertainties as stated in the caption. It seems to me that the color scales are not the correct ones. I apologize if I misunderstood something here.

We kept the color scale of the uncertainty figures a2 b2 c2 the same as the ones used for the best-fit result figures a1 b1 c1 to show the relatively small uncertainties we have for the wadsleyite proportion and ΔT . We now realize that this may cause some confusion and thus have revised the color scales of the uncertainty figures a2, b2 and c2 as suggested by the reviewer.

Supplementary information file

29) At line 13, Figure S1, I wonder if this spectrum should be labeled with Vs 1 + Vs 2. The full width at half maximum of the Vs peak is comparable to that of Vp. Due to the expected width scaling with frequency (or velocity) I would imagine this is a case of partial overlap of the two shear modes.

Thanks for pointing it out. We now marked Vs1 and Vs2 on this spectrum as shown on the bottom-left side of the figure below. We also included more spectra in the Figure S1, as shown below.

Fig. 7. Representative Brillouin spectra of single-crystal wadsleyite at different high P-T conditions. The spectra have been processed with a low pass filter. The collection time is approximately 20 minutes for each spectrum.

30) In Table 2, 3 and 4, the notation of the derivatives (in the columns titles) could be improved (see my comment 8, above)

We have updated all tables with the new notations.

31) In my opinion table 3 needs an additional note. “The entries at row 7 (reference 1) have been compared with an existing literature focusing on the pressure dependence of the elastic properties of wadsleyite at ambient temperature. The references to those studies can be found in Reference 1”. This cannot be overlooked, because these results (covering a range of wadsleyite compositions, i.e. H₂O contents and Fe contents) are part of the information which ultimately affects the modeling presented in this study, which includes the effect of H₂O content and of Fe content (which is neglected also based on the whole available set of previous studies).

Thank you. We have added this note in Supplementary Table 2 (old Table S3).

32) Equations S1 to S5 are missing in my version of the supplementary information file.

Sorry for the typo. we have changed “Equations S1-S5” to “Equations 1-5”, which refers to equations in the Methods section in the manuscript.

References:

- 1 Zhou, W.-Y. *et al.* The Water-Fe-Pressure dependent single-crystal elastic properties of wadsleyite: Implications for the seismic anisotropy in the upper Mantle Transition Zone. *Earth and Planetary Science Letters* **565**, 116955 (2021).
- 2 Libowitzky, E. & Rossman, G. R. An IR absorption calibration for water in minerals. *American Mineralogist* **82**, 1111-1115 (1997).
- 3 Buchen, J. *et al.* High-pressure single-crystal elasticity of wadsleyite and the seismic signature of water in the shallow transition zone. *Earth and Planetary Science Letters* **498**, 77-87, doi:10.1016/j.epsl.2018.06.027 (2018).
- 4 Mao, Z. *et al.* Single-crystal elasticity of wadsleyites, β -Mg₂SiO₄, containing 0.37–1.66 wt.% H₂O. *Earth and Planetary Science Letters* **268**, 540-549, doi:10.1016/j.epsl.2008.01.023 (2008).
- 5 Deon, F., Koch-Müller, M., Rhede, D. & Wirth, R. Water and Iron effect on the P-T-x coordinates of the 410-km discontinuity in the Earth upper mantle. *Contributions to Mineralogy and Petrology* **161**, 653-666, doi:10.1007/s00410-010-0555-6 (2010).
- 6 Chang, Y. Y. *et al.* Comparative compressibility of hydrous wadsleyite and ringwoodite: Effect of H₂O and implications for detecting water in the transition zone. *Journal of Geophysical Research: Solid Earth* **120**, 8259-8280, doi:10.1002/2015jb012123 (2015).
- 7 Jacobsen, S. D., Demouchy, S., Frost, D. J., Ballaran, T. B. & Kung, J. A systematic study of OH in hydrous wadsleyite from polarized FTIR spectroscopy and single-crystal X-ray diffraction: Oxygen sites for hydrogen storage in Earth’s interior. *American Mineralogist* **90**, 61-70 (2005).
- 8 Smyth, J. R., Bolfan-Casanova, N., Avignand, D., El-Ghozzi, M. & Hirner, S. M. Tetrahedral ferric iron in oxidized hydrous wadsleyite. *American Mineralogist* **99**, 458-466 (2014).

- 9 Buchen, J., Marquardt, H., Boffa Ballaran, T., Kawazoe, T. & McCammon, C. The equation of state of wadsleyite solid solutions: Constraining the effects of anisotropy and crystal chemistry. *American Mineralogist* **102**, 2494-2504, doi:10.2138/am-2017-6162 (2017).
- 10 Mao, Z. *et al.* Sound velocities of hydrous ringwoodite to 16 GPa and 673 K. *Earth and Planetary Science Letters* **331**, 112-119 (2012).
- 11 Schulze, K. *et al.* Seismically invisible water in Earth's transition zone? *Earth and Planetary Science Letters* **498**, 9-16 (2018).
- 12 <1984-sawamoto et al-brillouin-Mg100-ambient condition.pdf>.
- 13 Zha, C.-s. *et al.* Single-crystal elasticity of β -Mg₂SiO₄ to the pressure of the 410 km seismic discontinuity in the earth's mantle. *Earth and Planetary Science Letters* **147**, E9-E15 (1997).
- 14 Mao, Z. *et al.* Elasticity of hydrous wadsleyite to 12 GPa: Implications for Earth's transition zone. *Geophysical Research Letters* **35**, doi:10.1029/2008gl035618 (2008).
- 15 Mao, Z. *et al.* Effect of hydration on the single-crystal elasticity of Fe-bearing wadsleyite to 12 GPa. *American Mineralogist* **96**, 1606-1612, doi:10.2138/am.2011.3807 (2011).
- 16 Wang, J., Bass, J. D. & Kastura, T. Elastic properties of iron-bearing wadsleyite to 17.7 GPa: Implications for mantle mineral models. *Physics of the Earth and Planetary Interiors* **228**, 92-96 (2014).
- 17 Wang, W., Zhang, H., Brodholt, J. P. & Wu, Z. Elasticity of hydrous ringwoodite at mantle conditions: Implication for water distribution in the lowermost mantle transition zone. *Earth and Planetary Science Letters* **554**, 116626 (2021).
- 18 Ishii, T., Kojitani, H. & Akaogi, M. Phase relations and mineral chemistry in pyrolitic mantle at 1600–2200 °C under pressures up to the uppermost lower mantle: Phase transitions around the 660-km discontinuity and dynamics of upwelling hot plumes. *Physics of the Earth and Planetary Interiors* **274**, 127-137, doi:10.1016/j.pepi.2017.10.005 (2018).
- 19 Ishii, T., Kojitani, H. & Akaogi, M. Phase Relations of Harzburgite and MORB up to the Uppermost Lower Mantle Conditions: Precise Comparison With Pyrolite by Multisample Cell High-Pressure Experiments With Implication to Dynamics of Subducted Slabs. *Journal of Geophysical Research: Solid Earth* **124**, 3491-3507, doi:10.1029/2018jb016749 (2019).
- 20 Faul, U. H. & Jackson, I. The seismological signature of temperature and grain size variations in the upper mantle. *Earth and Planetary Science Letters* **234**, 119-134 (2005).
- 21 Karato, S.-i. On the origin of the asthenosphere. *Earth and Planetary Science Letters* **321**, 95-103 (2012).
- 22 Koelemeijer, P., Ritsema, J., Deuss, A. & van Heijst, H. J. SP12RTS: a degree-12 model of shear- and compressional-wave velocity for Earth's mantle. *Geophysical Journal International* **204**, 1024-1039, doi:10.1093/gji/ggv481 (2015).
- 23 Liu, J., Chen, G., Gwanmesia, G. D. & Liebermann, R. C. Elastic wave velocities of pyrope–majorite garnets (Py62Mj38 and Py50Mj50) to 9 GPa. *Physics of the Earth and Planetary Interiors* **120**, 153-163 (2000).
- 24 Sinogeikin, S. V. & Bass, J. D. Elasticity of pyrope and majorite–pyrope solid solutions to high temperatures. *Earth and Planetary Science Letters* **203**, 549-555 (2002).
- 25 Irifune, T. *et al.* Sound velocities of majorite garnet and the composition of the mantle transition region. *Nature* **451**, 814-817, doi:10.1038/nature06551 (2008).
- 26 Liu, Z. *et al.* Influence of aluminum on the elasticity of majorite-pyrope garnets. *American Mineralogist: Journal of Earth and Planetary Materials* **104**, 929-935 (2019).
- 27 Arimoto, T., Gréaux, S., Irifune, T., Zhou, C. & Higo, Y. Sound velocities of Fe₃Al₂Si₃O₁₂ almandine up to 19 GPa and 1700 K. *Physics of the Earth and Planetary Interiors* **246**, 1-8, doi:10.1016/j.pepi.2015.06.004 (2015).

- 28 Gwanmesia, G. D., Wang, L., Heady, A. & Liebermann, R. C. Elasticity and sound velocities of polycrystalline grossular garnet (Ca₃Al₂Si₃O₁₂) at simultaneous high pressures and high temperatures. *Physics of the Earth and Planetary Interiors* **228**, 80-87 (2014).
- 29 Zhou, C. *et al.* Sound velocity of MgSiO₃ majorite garnet up to 18 GPa and 2000 K. *Geophysical Research Letters* **48**, e2021GL093499 (2021).
- 30 Liu, Z. *et al.* Elastic wave velocity of polycrystalline Mj 80 Py 20 garnet to 21 GPa and 2,000 K. *Physics and Chemistry of Minerals* **42**, 213-222 (2015).
- 31 Irifune, T. *et al.* Sound velocities of majorite garnet and the composition of the mantle transition region. *Nature* **451**, 814-817 (2008).
- 32 Kono, Y., Gréaux, S., Higo, Y., Ohfuji, H. & Irifune, T. Pressure and temperature dependences of elastic properties of grossular garnet up to 17 GPa and 1 650 K. *Journal of Earth Science* **21**, 782-791 (2010).
- 33 Huang, Q., Schmerr, N., Waszek, L. & Beghein, C. Constraints on Seismic Anisotropy in the Mantle Transition Zone From Long-Period SS Precursors. *Journal of Geophysical Research: Solid Earth* **124**, 6779-6800, doi:10.1029/2019jb017307 (2019).
- 34 Ritsema, J., Deuss, a. A., Van Heijst, H. & Woodhouse, J. S40RTS: a degree-40 shear-velocity model for the mantle from new Rayleigh wave dispersion, teleseismic traveltime and normal-mode splitting function measurements. *Geophysical Journal International* **184**, 1223-1236 (2011).

REVIEWERS' COMMENTS

Reviewer #1 (Remarks to the Author):

The revised manuscript has appropriately addressed all my concerns. I would like to recommend the publication of the manuscript in Nature Communications.

Reviewer #2 (Remarks to the Author):

The authors have provided an exceptionally thorough response to the comments.

The question of chemical variation (i.e. lateral change in chemistry) is not fully addressed on Lines 120-124. Varying the ratio of wadsleyite/majorite does of course vary the chemical composition of the transition zone, but this does not cover the full range of compositional variation that is possible. It represents only one type of compositional change in which various components are correlated. A much wider range of compositions is certainly possible and the authors should at least acknowledge this fact. That said, I think the paper does represent a step forward in upper mantle modeling and should be published.

Response to reviewers' comments on manuscript NCOMMS-21-26303-T: "Constraining Composition and Temperature Variations in the Mantle Transition Zone" by Wen-Yi Zhou, Ming Hao, Jin S. Zhang, Bin Chen, Ruijia Wang, and Brandon Schmandt.

We sincerely thank the editor and all reviewers for all the help during the revision process.

Response to reviewer 1

The revised manuscript has appropriately addressed all my concerns. I would like to recommend the publication of the manuscript in Nature Communications.

We thank the reviewer 1 for the positive evaluation of the revised manuscript.

Response to reviewer 2

The authors have provided an exceptionally thorough response to the comments.

The question of chemical variation (i.e. lateral change in chemistry) is not fully addressed on Lines 120-124. Varying the ratio of wadsleyite/majorite does of course vary the chemical composition of the transition zone, but this does not cover the full range of compositional variation that is possible. It represents only one type of compositional change in which various components are correlated. A much wider range of compositions is certainly possible and the authors should at least acknowledge this fact. That said, I think the paper does represent a step forward in upper mantle modeling and should be published.

We thank the reviewer 2 for the positive evaluation of the revised manuscript and insightful suggestion. We agree with the reviewer 2 and have added some sentences in the main text to point out this limitation as suggested by reviewer 2 (Lines 123-125).